# CAUSAL REINFORCEMENT LEARNING USING OBSERVATIONAL AND INTERVENTIONAL DATA

## ABSTRACT

Learning efficiently a causal model of the environment is a key challenge of model-based RL agents operating in POMDPs. We consider here a scenario where the learning agent has the ability to collect online experiences through direct interactions with the environment (interventional data), but also has access to a large collection of offline experiences, obtained by observing another agent interacting with the environment (observational data). A key ingredient, which makes this situation non-trivial, is that we allow the observed agent to act based on privileged information, hidden from the learning agent. We then ask the following questions: can the online and offline experiences be safely combined for learning a causal transition model ? And can we expect the offline experiences to improve the agent's performances ? To answer these, first we bridge the fields of reinforcement learning and causality, by importing ideas from the well-established causal framework of do-calculus, and expressing model-based reinforcement learning as a causal inference problem. Second, we propose a general yet simple methodology for safely leveraging offline data during learning. In a nutshell, our method relies on learning a latent-based causal transition model that explains both the interventional and observational regimes, and then inferring the standard POMDP transition model via deconfounding using the recovered latent variable. We prove our method is correct and efficient in the sense that it attains better generalization guarantees due to the offline data (in the asymptotic case), and we assess its effectiveness empirically on a series of synthetic toy problems.

## 1 INTRODUCTION

As human beings, a key ingredient in our learning process is experimentation: we perform actions in our environment and we measure their outcomes. Another ingredient, maybe less understood, is observation: we observe the behaviour of others acting and evolving in the environment, be it people, animals, or even plants. It is well-known that observation alone is not sufficient to infer how our environment works, or more precisely to predict the outcome of our actions, especially when the behaviours we observe depend on hidden information[1]. And yet a whole field of science, astronomy, heavily relies on the observation of celestial bodies in the sky, on which experimentation is virtually impossible. So which role exactly does observation play during learning ? And in particular, how do we combine observation and experimentation ?

In the context of reinforcement learning (RL), a related question is the combination of offline data, resulting from observations, with online data resulting from experimentation, in order to improve the performance of a learning agent. In the Markov Decision Process (MDP) setting, where the agent observes the entire state of the environment, the answer is straightforward and practical solutions exist, leading to the fastly growing field of offline reinforcement learning [17; 18] where large databases of demonstrations can be efficiently leveraged. In the more general Partially-Observable MDP (POMDP) setting however, the question turns out to be much more challenging. A typical example is in the context of medicine, where offline data is collected from physicians who may rely on information absent from their patient's medical records, such as their wealthiness or their lifestyle. Suppose that wealthy patients in general get prescribed specific treatments by their physicians, because they can

---

[1]Simply put, correlation does not imply causation. Or, citing Pearl [24], "behind every causal conclusion there must lie some causal assumption that is not testable in observational studies".

afford it, while being less at risk to develop severe conditions regardless of their treatment, because they can also afford a healthier lifestyle. This creates a spurious correlation called confounding, and will cause a naive recommender system to wrongly infer that a treatment has positive health effects. A second example is in the context of autonomous driving, where offline data is collected from human drivers who have a wider field of vision than the camera on which the robot driver relies. Suppose human drivers push the brakes when they see a person waiting to cross the street, and only when the person walks in front of the car it enters the camera's field of vision. Then, again, a naive robot might wrongly infer from its observations that whenever brakes are pushed, a person appears in front of the car. Suppose now that the robot's objective is to avoid collisions with pedestrians, they it might get regrettably reluctant to push the brakes. Of course, in both those situations, the learning agent will eventually infer the right causal effects of its actions if it collects enough online data from its own interactions. However, in both those situations also, performing many interventions for the sole purpose of seeing what happens is not really realistic, while collecting offline data by observing the behaviour of human agents is much more affordable.

In this paper we study the question of combining offline and online data under the Partially-Observable Markov Decision Process (POMDP) setting, by importing tools and ideas from the well-established field of causality [23] into the model-based RL framework. Our contribution is three-fold:

1. We formalize model-based RL as a causal inference problem using the framework of *do*-calculus [25], which allows us to reason formally about online and offline scenarios in an intuitive manner (Section 3).

2. We present a generic method for combining offline and online data in model-based RL (Section 4), with a formal proof of correctness even when the offline policy relies on privileged hidden information (confounding variable), and a proof of efficiency in the asymptotic case (with respect to using online data only).

3. We propose a practical implementation of our method, and illustrate its effectiveness on three experiments with synthetic toy problems (Section 6).

While our proposed method can be formulated outside of the framework of *do*-calculus, in this paper we hope to demonstrate that it offers a principled and intuitive tool to reason about model-based RL. By relating common concepts from RL and causality, we wish that our contribution will ultimately help to bridge the gap between the two communities.

## 2 BACKGROUND

### 2.1 NOTATION

In this paper, upper-case letters in italics denote random variables (e.g. $X, Y$), while their lower-case counterpart denote their value (e.g. $x, y$) and their calligraphic counterpart their domain (e.g., $x \in \mathcal{X}$). We consider only discrete random variables. To keep our notation uncluttered, with a slight abuse of notations and use $p(x)$ to denote sometimes the event probability $p(X = x)$, and sometimes the whole probability distribution of $X$, which should be clear from the context. In the context of sequential models we also distinguish random variables with a temporal index $t$, which might be fixed (e.g., $o_0, o_1$ ), or undefined (e.g., $p(s_{t+1}|s_t, a_t)$ denotes at the same time the distributions $p(s_1|s_0, a_0)$ and $p(s_2|s_1, a_1)$). We also adopt a compact notation for sequences of contiguous variables (e.g., $s_{0 \to T} = (s_0, \dots, s_T) \in \mathcal{S}^{T+1}$ ), and for summations over sets ($\sum_{x \in \mathcal{X}} \iff \sum_x^{\mathcal{X}}$). We assume the reader is familiar with the concepts of conditional independence ($X \perp\!\!\!\perp Y \mid Z$) and probabilistic graphical models based on directed acyclic graphs (DAGs), which can be found in most introductory textbooks, e.g. Pearl [22]; Studeny [29]; Koller and Friedman [15]. In the following we will use *do*-calculus to derive formal solutions to model-based RL in various POMDP settings. We refer the reader to Pearl [25] for a thorough introduction, and give a description of rules R1, R2 and R3 used in our derivations in the appendix (Section A.1).

### 2.2 PARTIALLY-OBSERVABLE MARKOV DECISION PROCESS

We consider Partially-Observable Markov Decision Processes (POMDPs) of the form $M = (\mathcal{S}, \mathcal{O}, \mathcal{A}, p_{init}, p_{obs}, p_{trans}, r)$, with hidden states $s \in \mathcal{S}$, observations $o \in \mathcal{O}$, actions $a \in \mathcal{A}$, ini-

tial state distribution $p_{init}(s_0)$, state transition distribution $p_{trans}(s_{t+1}|s_t, a_t)$, observation distribution $p_{obs}(o_t|s_t)$, and reward[2] function $r : \mathcal{O} \to \mathbb{R}$. For simplicity we assume episodic tasks with finite horizon $H$. We further denote a complete trajectory $\tau = (o_0, a_0, \dots, o_H)$, and for convenience we introduce the concept of a history at time $t$, $h_t = (o_0, a_0, \dots, o_t)$.

A common control scenario for POMDPs is when actions are decided based on all the available information from the past. We call this the *standard POMDP setting*. The control mechanism can be represented as a stochastic policy $\pi(a_t|h_t)$, which together with the POMDP dynamics $p_{init}$, $p_{obs}$ and $p_{trans}$ defines a probability distribution over trajectories $\tau$,

$$p_{std}(\tau) = \sum_{s_{0 \to |\tau|}}^{\mathcal{S}^{|\tau|+1}} p_{init}(s_0)p_{obs}(o_0|s_0) \prod_{t=0}^{|\tau|-1} \pi(a_t|h_t)p_{trans}(s_{t+1}|s_t, a_t)p_{obs}(o_{t+1}|s_{t+1}).$$

This whole data-generation mechanism can be represented visually as a DAG, represented in Figure 1. A key characteristic in this setting is that $A_t \perp\!\!\!\perp S_t \mid H_t$ is always true, that is, every action is independent of the current state given the history.

## 2.3 MODEL-BASED RL

Assuming the objective is the long-term reward, the POMDP control problem formulates as:

$$\pi^\star = \arg\max_\pi \ \mathbb{E}_{\tau \sim p_{std}} \left[ \sum_{t=0}^{|\tau|} r(o_t) \right]. \tag{1}$$

Model-based RL relies on the estimation of the POMDP transition model $p_{std}(o_{t+1}|h_t, a_t)$ to solve (1), which decomposes into two sub-problems:

1. learning: given a dataset $\mathcal{D}$, estimate a transition model $\hat{q}(o_{t+1}|h_t, a_t) \approx p_{std}(o_{t+1}|h_t, a_t)$;

2. planning: given a history $h_t$ and a transition model $\hat{q}$, decide on an optimal action $a_t$.

As we will see shortly, the transition model $\hat{q}$ sought by model-based RL is inherently causal [9]. In this work we consider only the first problem above, that is, learning the (causal) POMDP transition model from data.

## 3 MODEL-BASED RL AS CAUSAL INFERENCE

Decision problems, such as those arising in POMDPs, can naturally be formulated in terms of causal queries where actions directly translate into $do$ statements. For example, given past information about the POMDP process, what will be the causal effect of an action (intervention) on future rewards ? [3]

### 3.1 THE INTERVENTIONAL REGIME

In the interventional regime, we assume a dataset $\mathcal{D}_{int}$ of episodes $\tau$ collected in the standard POMDP setting from an arbitrary decision policy $\pi(a_t|h_t)$,

$$\mathcal{D}_{int} \sim p_{init}, p_{trans}, p_{obs}, \pi.$$

Let us now adopt a causal perspective and reason in terms of interventions in the causal system, depicted in Figure 1. Consider that we want to control the system, that is, replace $\pi$ with $\pi^\star$, in order to maximize a long-term outcome. Then, evaluating the effect of each action on the system is a causal inference problem. In order to decide on the best first action $a_0$ given $h_0 = (o_0)$, one must evaluate a series of causal queries in the form $p_{std}(o_1|o_0, do(a_0))$, then $p_{std}(o_2|o_0, do(a_0), o_1, do(a_1))$, and so on, and finally using those causal distributions for planning by solving a Bellman equation. Conveniently, in the interventional regime, applying rule R2 of do-calculus on the causal DAG results

---

[2]Without loss of generality we consider the reward to be part of the observation $o_t$ to simplify our notation.
[3]A guiding example accompanying this section can be found in the appendix (Section A.3).

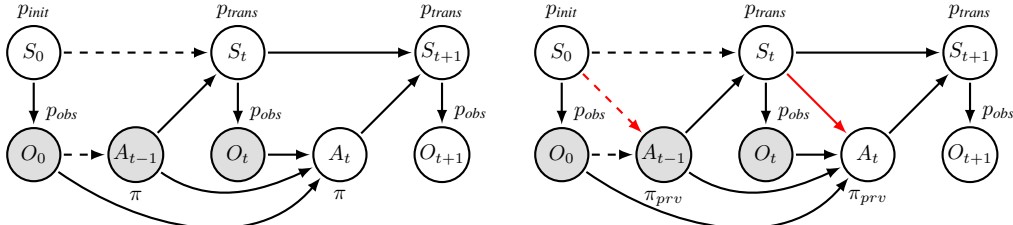

Figure 1: Standard POMDP setting.      Figure 2: Privileged POMDP setting.

in those queries being trivially identifiable from $p_{std}(\tau)$. In fact, those queries exactly boil down to the standard POMDP transition model that model-based RL seeks to estimate,

$$p_{std}(o_{t+1}|o_{0 \to t}, do(a_{0 \to t})) = p_{std}(o_{t+1}|h_t, a_t). \quad (2)$$

As such, model-based RL can be naturally reinterpreted in terms of causal inference. Also, a convenient property in this regime is that $p_{std}(o_{t+1}|h_t, a_t)$ does not depend on the control policy $\pi$ that was used to build the dataset $\mathcal{D}_{int}$. The only requirement, in order to estimate transition probabilities for every $h_t, a_t$ combination, is that $\pi$ has a non-zero chance to explore every action, that is, $\pi(a_t|h_t) > 0, \forall a_t, h_t$. Then, an unbiased estimate of the standard POMDP transition model can be obtained simply via log-likelihood maximization:

$$\hat{q} = \arg\max_{q \in \mathcal{Q}} \sum_{\tau}^{\mathcal{D}_{int}} \sum_{t=0}^{|\tau|-1} \log q(o_{t+1}|h_t, a_t). \quad (3)$$

In some situations it is very reasonable to assume an interventional regime, for example when it is known to hold by construction. This is the case with online RL data, as the learning agent itself explicitly controls the data-collection policy $\pi(a_t|h_t)$. But it can also be the case with offline RL data, if one knows that the data-collection policy did not use any additional information besides the information available to the learning agent, $h_t$. In Atari video games for example, it is hard to imagine a human player using any kind of privileged information related to the machine's internal state $s_t$ other than the video and audio outputs from the game.

## 3.2 THE OBSERVATIONAL REGIME

In the observational regime, we assume a dataset $\mathcal{D}_{obs}$ of episodes $\tau$ collected in the *privileged POMDP setting*, depicted in Figure 2. In this setting episodes are collected from an external agent who has access to privileged information, in the extreme case the whole POMDP state $s_t$, which the learning agent can not observe[4]. In this setting we denote the data-generating control policy $\pi_{prv}(a_t|h_t, s_t)$, such that

$$\mathcal{D}_{obs} \sim p_{init}, p_{trans}, p_{obs}, \pi_{prv}.$$

We denote the whole episode distribution resulting from $p_{init}, p_{trans}, p_{obs}$ and $\pi_{prv}$ as $p_{prv}(\tau)$. A key characteristic in this setting is that now $A_t \perp\!\!\!\perp S_t \mid H_t$ can not be assumed to hold any more.

Let us reason here again in terms of causal inference from the causal system depicted in Figure 2. For the purpose of controlling the POMDP in the standard setting, in the light of past information $h_t$, we want to evaluate the same series of causal queries as before, in the form $p_{prv}(o_{t+1}|o_{0 \to t}, do(a_{0 \to t}))$. This time however, those causal queries are not identifiable from $p_{prv}(\tau)$. Evaluating them would require knowledge of the POMDP hidden states $s_t$, which act as confounding variables. For example, identifying the first query at $t = 0$ requires at least the observation of $s_0$,

$$p_{prv}(o_1|o_0, do(a_0)) = \sum_{s_0 \in \mathcal{S}} p_{prv}(s_0|o_0, do(a_0)) p_{prv}(o_1|s_0, o_0, do(a_0))$$

$$= \sum_{s_0 \in \mathcal{S}} p_{prv}(s_0|o_0) p_{prv}(o_1|s_0, a_0)$$

---

[4]Note that our only assumption is that this external agent has access to privileged information. We do not assume it acts optimally with respect to the learning agent's reward, or any other reward.

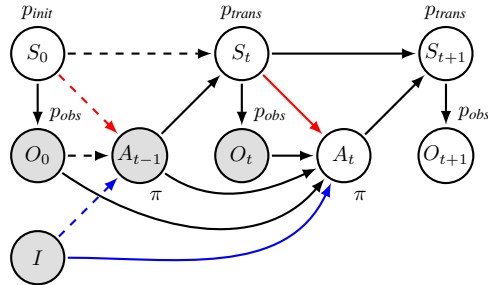

Figure 3: Augmented POMDP setting, with a policy regime indicator $I$ taking values in $\{0, 1\}$ (1=interventional regime, no confounding, 0=observational regime, potential confounding), such that $\pi(a_t|h_t, s_t, i = 1) = \pi(a_t|h_t, i = 1)$. This additional constraint introduces a contextual independence $A_t \perp\!\!\!\perp S_t \mid H_t, I = 1$.

(R3 and R2 of do-calculus, then $O_{t+1} \perp\!\!\!\perp H_t \mid S_t, A_t$).

In many offline RL situations, we believe that it is common to have access to POMDP trajectories for which $A_t \perp\!\!\!\perp S_t \mid H_t$ can not be assumed, for example when demonstrations are collected from a human agent acting in the world (see Section 1 for examples). In such a situation, the observed trajectories may be confounded, and naively learning a causal transition model by solving (3) might result in a non-causal model, and in non-optimal planning. A natural question is then: what can be done in such a situation ? Are confounded trajectories useless ? Is there still a way to use this data ?

## 4 COMBINING OBSERVATIONAL AND INTERVENTIONAL DATA

### 4.1 PROBLEM STATEMENT

We consider a generic situation where two datasets of POMDP trajectories $\mathcal{D}_{int}$ and $\mathcal{D}_{obs}$ are available, sampled respectively in the interventional regime with policy $\pi_{std}(a_t|h_t)$, and in the observational (potentially confounded) regime with policy $\pi_{prv}(a_t|h_t, s_t)$. We then ask the following question: is there a sound way to use the observational data for improving the estimator of the standard POMDP transition model that would be recovered from the interventional data only ?

### 4.2 THE AUGMENTED POMDP

We formulate the problem of learning the standard POMDP transition model from $\mathcal{D}_{int}$ and $\mathcal{D}_{obs}$ as that of inferring a structured latent-variable model. Since both datasets are sampled from the same POMDP ($p_{init}$, $p_{trans}$ and $p_{obs}$) controlled in different ways (either $\pi_{prv}$ or $\pi_{std}$), the overall data generating process can be represented in the form of an augmented DAG, depicted in Figure 3. We simply introduce an auxiliary variable $I \in \{0, 1\}$ that acts as a regime indicator [5], for differentiating between observational and interventional data. The augmented POMDP policy then simply becomes $\pi$, where $\pi(a_t|h_t, s_t, i = 0) = \pi_{prv}(a_t|h_t, s_t)$ and $\pi(a_t|h_t, s_t, i = 1) = \pi_{std}(a_t|h_t)$.

For simplicity, in the following we will refer to the joint distribution of this augmented POMDP as the true distribution $p$, and with a slight abuse of notation we will consider $\mathcal{D}_{obs}$ and $\mathcal{D}_{int}$ two datasets of augmented POMDP trajectories, sampled respectively under the observational regime $(\tau, i) \sim p(\tau, i|i = 0)$, and the interventional regime $(\tau, i) \sim p(\tau, i|i = 1)$. The causal queries required to control the augmented POMDP can then be identified as

$$p(o_{t+1}|o_{0 \to t}, do(a_{0 \to t})) = p(o_{t+1}|o_{0 \to t}, do(a_{0 \to t}), i = 1)$$
$$= p(o_{t+1}|h_t, a_t, i = 1)$$

(R1 of do-calculus, then R2 on the contextual causal DAG from Figure 1).

### 4.3 THE AUGMENTED LEARNING PROBLEM

In order to learn the standard POMDP transition model $p(o_{t+1}|h_t, a_t, i = 1)$ from the augmented dataset $\mathcal{D}_{obs} \cup \mathcal{D}_{int} = \mathcal{D} \sim p(\tau, i)$, we propose the following two-step procedure.

**Learning** In the first step, we fit a latent probabilistic model $\hat{q}$ to the training trajectories, constrained to respect all the independencies of our augmented POMDP. To do so we substitute the actual POMDP hidden state $s_t \in \mathcal{S}$ by a latent variable $z_t \in \mathcal{Z}$, with $\mathcal{Z}$ the discrete latent space of the model. Our learning problem then formulates as a standard likelihood maximization[5], i.e.,

$$\hat{q} = \arg\max_{q \in \mathcal{Q}} \sum_{(\tau, i)}^{\mathcal{D}} \log q(\tau, i), \tag{4}$$

with $\mathcal{Q}$ the family of sequential latent probabilistic models that respect

$$q(\tau, i) = q(i) \sum_{z_{0 \to |\tau|}}^{\mathcal{Z}^{|\tau|+1}} q(z_0) q(o_0 | z_0) \prod_{t=0}^{|\tau|-1} q(a_t | h_t, z_t, i) q(z_{t+1} | a_t, z_t) q(o_{t+1} | z_{t+1}),$$

$$q(a_t | h_t, z_t, i = 1) = q(a_t | h_t, i = 1).$$

**Inference** In the second step, we recover $\hat{q}(o_{t+1} | h_t, a_t, i = 1)$ as an estimator of the standard POMDP transition model. This can be done efficiently with a forward algorithm over the augmented DAG structure[6], which unrolls over time as the RL agent evolves in the environment.

Intuitively, the observational data $\mathcal{D}_{obs}$ influences the interventional transition model $q(o_{t+1} | h_t, a_t, i = 1)$ as follows. The learned model $q$ must fit the observational and interventional data by sharing the same building blocs $q(z_0)$, $q(o_t | z_t)$ and $q(z_{t+1} | z_t, a_t)$, while only the expert policy $q(a_t | h_t, z_t, i = 0)$ offers some flexibility that allows to differentiate between both regimes. As a result, imposing an observational distribution $q(\tau | i = 0)$ acts as a regularizer for the interventional distribution $q(\tau | i = 1)$.

## 4.4 THEORETICAL GUARANTEES

In this section we show that our two-step approach is 1) correct, in the sense that it yields an unbiased estimator of the standard POMDP causal transition model and 2) efficient, in the sense that it yields a better estimator than the one based on interventional data only (asymptotically in the number of observational data). All proofs are deferred to the appendix (Section A.7).

First we show that the recovered estimator is unbiased, and then we derive bounds for $\hat{q}(o_{t+1} | h_t, a_t, i = 1)$ in the asymptotic observational scenario, $|\mathcal{D}_{obs}| \to \infty$ (regardless of the interventional data $\mathcal{D}_{int}$).

**Proposition 1.** *Assuming $|\mathcal{Z}| \geq |\mathcal{S}|$, $\hat{q}(o_{t+1} | h_t, a_t, i = 1)$ is an unbiased estimator of $p(o_{t+1} | h_t, a_t, i = 1)$.*

**Theorem 1.** *Assuming $|\mathcal{D}_{obs}| \to \infty$, for any $\mathcal{D}_{int}$ the recovered causal model is bounded as follows:*

$$\prod_{t=0}^{T-1} \hat{q}(o_{t+1} | h_t, a_t, i = 1) \geq \prod_{t=0}^{T-1} p(a_t | h_t, i = 0) p(o_{t+1} | h_t, a_t, i = 0), \text{ and}$$

$$\prod_{t=0}^{T-1} \hat{q}(o_{t+1} | h_t, a_t, i = 1) \leq \prod_{t=0}^{T-1} p(a_t | h_t, i = 0) p(o_{t+1} | h_t, a_t, i = 0) + 1 - \prod_{t=0}^{T-1} p(a_t | h_t, i = 0),$$

*$\forall h_{T-1}, a_{T-1}, T \geq 1$ where $p(h_{T-1}, a_{T-1}, i = 0) > 0$.*

As a direct consequence, in the asymptotic case, using observational data ensures stronger generalization guarantees for the recovered transition model than using no observational data.

**Corollary 1.** *The estimator $\hat{q}(o_{t+1} | h_t, a_t, i = 1)$ recovered after solving (4) with $|\mathcal{D}_{obs}| \to \infty$ offers strictly better generalization guarantees than the one with $|\mathcal{D}_{obs}| = 0$, for any $\mathcal{D}_{int}$.*

---

[5]Note that, while the problem of learning structured latent variable models is known to be hard in general, there also exists a wide range of tools and algorithms available in the literature for solving it approximately, such as the EM algorithm or the method of ELBO maximization.

[6]See appendix for details (Section A.4)

## 5 RELATED WORK

A whole body of work exists around the question of merging interventional and observational data in RL, with related results already in econometrics [20]. Bareinboim et al. [2] study a sequential decision problem similar to ours, but assume that expert intentions are observed both in the interventional and the observational regimes, i.e., prior to doing interventions the learning agent can ask "what would the expert do in my situation ?" This introduces an intermediate, observed variable $\hat{a}_t = f(o_t)$ with the property that $p_{prv}(a_t = \hat{a}_t | \hat{a}_t) = 1$, which guarantees unconfoundedness in the observational regime ($A_t \perp\!\!\!\perp S_t | H_t$), so that observational data can be considered interventional, and the standard PO-MDP transition model can be directly estimated via (3). Zhang and Bareinboim [31; 34] relax this assumption in the context of binary bandits, and later on in the more general context of dynamic treatment regimes [32; 33]. They derive causal bounds similar to ours (Theorem 1), and propose a two-step approach which first extracts causal bounds from observational data, and then uses these bounds in an online RL algorithm. While their method nicely tackles the question of leveraging observational data for online exploration, it does not account for uncertainty in the bounds estimated from the observational data. In comparison, our latent-based approach is more flexible, as it never computes explicit bounds, but rather lets the learning agent decide through (4) how data from both regimes influence the final transition model, depending of the number of samples available. Kallus et al. [13] also propose a two-step learning procedure to combine observational and interventional data in the context of binary contextual bandits. Their method however relies on a series of strong parametric assumptions (strong one-way overlap, linearity, non-singularity etc.).

A specific instantiation of our framework is off-policy evaluation, i.e., estimating the performance of a policy $\pi$ using observational data only. This corresponds to the specific setting $|\mathcal{D}_{int}| = 0$, where it can be shown that the causal transition model is in general not recoverable in the presence of confounding variables. Still, a growing body of literature studies the question under specific structural or parametric assumptions [19; 30; 3]. In the context of imitation learning, de Haan et al. [6] attribute the issue of *causal misidentification*, that is, ascribing the actions of an agent to the wrong explanatory variables, to confounding. We argue that this explanation is erroneous, since their imitated experts are trained in the standard POMDP setting (interventional regime). This reasoning supports Spencer et al. [28], who shows that *causal misidentification* is simply a manifestation of *covariate shift*. Finally, other issues orthogonal to confounding can appear when combining online and offline data RL, for example the value function initialisation problem [8], or the bootstrapping error problem [16; 21].

## 6 EXPERIMENTS

We perform experiments on three synthetic toy problems, each one expressing a different level of complexity and a different form of hidden information.

**Door** In this toy problem, we consider a closed door, and a light (red or green) indicating which of two buttons (A and B) should be pressed to open the door. The privileged agent perceives the color of the light, while the learning agent doesn't (colorblind). This corresponds to a simple binary bandit, with a time horizon $H = 1$ and a hidden state space $|\mathcal{S}| = 3$.

**Tiger** In this classical problem from the literature [4], the agent stands in front of two doors, one with a treasure behind (+10 reward) and one with a tiger behind (-100 reward). At each time step the agent can either open one of the doors, or listen (-1 reward) to obtain a noisy estimate of the tiger's position. The privileged agent has full knowledge of the tiger's position, while the learning agent doesn't. This toy problem is a small-scale POMDP, with a time horizon $H = 50$ and a hidden state space $|\mathcal{S}| = 6$.

**Gridworld** This problem is inspired from Alt et al. [1]. Here the agent starts on the top-left corner of a small 5x5 grid, and tries to get to a target placed on the bottom side behind a large wall. The agent can use five actions: *top*, *right*, *bottom*, *left* and *idle*, and moves into the desired direction with 50% chances, or randomly remains in the current tile or slips to one of the 4 adjacent tiles otherwise. The privileged agent has full knowledge of its position at each time step, while the learning agent is only revealed this information once in a while, with 20% chances. This toy problem constitutes a more challenging POMDP, with a time horizon $H = 20$ and a hidden state space $|\mathcal{S}| = 42$.

For each toy problem, we train and evaluate our proposed approach, *augmented*, using a large amount of observational data $\mathcal{D}_{obs}$ (512 samples for *door*, 8192 for *tiger* and *gridworld*), and interventional

data $\mathcal{D}_{int}$ of varying size, collected from random explorations. Each time, we compare our approach to two baseline methods: *no obs*, where the $\mathcal{D}_{obs}$ is not used at all during training, and *naive*, where $\mathcal{D}_{obs}$ is naively combined with $\mathcal{D}_{int}$ as if there was no confounding. As a reference, in the *door* experiment we also report the performance of Kallus et al. [13] (only setting in which it applies). We repeat each experiment over 20 random seeds. In the following we report our main results, and defer the reader to the appendix (Section A.5) for the complete experimental details and results. [7]

## 6.1 PERFORMANCE OF THE RL AGENTS

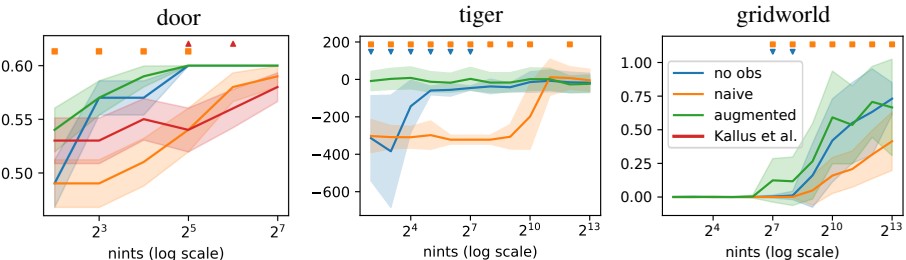

Figure 4: Performance of each RL agent on our three toy problems (the higher the better). We report the average cumulative reward (mean $\pm$ std) obtained on the real environment. Little markers indicate the significance of a two-sided Wilcoxon signed-rank test [7] with $\alpha < 5\%$, between our method, *augmented*, and the baselines *no obs* (down triangles), *naive* (squares) and Kallus et al. (up triangles).

In Figure 4 we report the test performance of an RL agent trained on the transition models recovered by each method. Here the privileged policy $\pi_{prv}$ consists of a good but imperfect expert in both *door* and *tiger*, and a shortest-path expert in *gridworld*. In all three toy problems, our method successfully leverages the confounded observational data and outperforms the two baseline methods, especially in the low-sample regime (few interventional samples). Most noticeably, the *no obs* baseline converges to the same performance as our method, if given enough interventional samples, while the *naive* baseline seems to suffer from the additional observational data, and converges much slower than the two other methods. As a reference, our approach also performs much better than Kallus et al. [13].

## 6.2 ROBUSTNESS TO DIFFERENT DEGREES OF CONFOUNDING

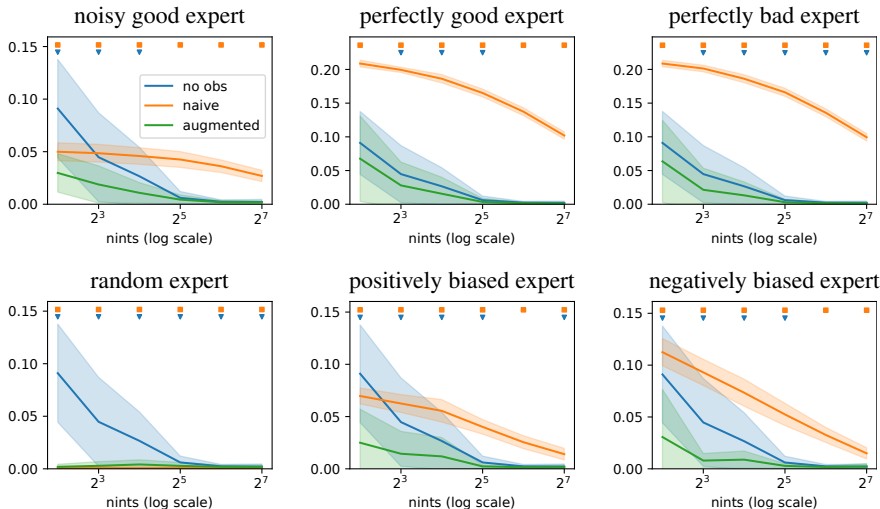

Figure 5: Robustness to different degrees of confounding in the *door* problem (the lower the better). We report the JS divergence (mean $\pm$ std) between the true and the recovered transition model.

---

[7]Code available at `https://github.com/causal-rl-anonymous/causal-rl`

Our approach is robust to any kind of confounding, and does not assume the observed expert uses the privileged information in any specific way, or performs well or poorly at the task at hand. To empirically demonstrate this claim, we repeat the *door* experiment with various expert behaviours, including perfectly good / bad (always / never press the correct button), positively / negatively biased (overly optimistic / pessimistic towards the optimal button), and random as a control (no confounding). The outcome of this experiment is showcased in Figure 5, where our method always results in the best estimate of the transition model (except in the situation with no confounding, where the *naive* approach is slightly more effective). In the appendix (Section A.6.1) we also report gains in terms of reward, except when there is no confounding or when the confounding induces a positive bias, in which case the *naive* approach performs slightly better.

## 6.3 FOCUS ON THE GRIDWORLD PROBLEM

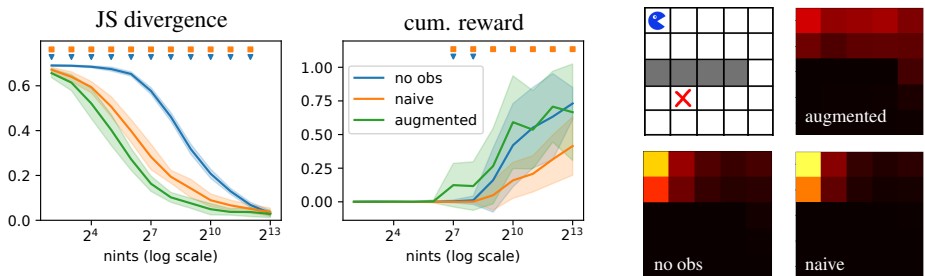

Figure 6: The gridworld experiment. **Left**: the JS divergence and cumulative reward obtained by each method. **Right**: the initial grid, and a heatmap of the tiles visited by the RL agents at test time at the $|\mathcal{D}_{int}| = 2^7$ mark. At this point, only the *augmented* method has learned how to pass the wall.

Let us now have a particular focus on our most complex problem, the *gridworld* problem, showcased in Figure 6. Our method, *augmented*, starts obtaining positive rewards with $2^7 = 128$ interventional samples, which is two orders of magnitude better than the two other methods, where positive rewards appear later on at $2^9 = 512$ samples. This impact is clearly noticeable if one looks at the test-time trajectories of the RL agents at the $2^7$ mark (Figure 6, right side). While the transition models learned by the *no obs* and *naive* approaches result in the agent being stuck around its starting position, the transition model learned by our approach already allows the agent to learn how to pass the wall and reach the bottom side of the grid.

## 7 DISCUSSIONS

In this paper we have presented a simple, generic method for combining interventional and observational (potentially confounded) data in model-based reinforcement learning for POMDPs. We have demonstrated that our method is correct and efficient in the asymptotic case (infinite observational data), and we have illustrated its effectiveness on three synthetic toy problems. One limitation of our method is that it adds an additional challenge on top of model-based RL, that of learning a latent-based transition model, which can become problematic in high-dimensional RL settings.. Still, the recent success of discrete latent models for solving complex RL tasks [10] or tasks in high-dimensional domains [26] lets us envision that this difficulty can be overcome in practice. A first potential extension to our work could be to use offline data to guide online exploration, in a fashion similar to Zhang and Bareinboim [31; 32; 33; 34]. A second direct extension is to consider that several agents are observed, each with its own privileged policy, leading to multiple observational regimes. This would lead, in the asymptotic case, to a stronger implicit regularizer for the causal transition model. A third, obvious extension is to develop a similar approach for model-free RL, maybe in the form of a value-function regularizer. A fourth direction is to apply the same approach to POMDPs with continuous observation spaces (e.g., pixel-based problems), which is theoretically very straightforward. Finally, we hope that our work will help to bridge the gap between the RL and causality communities, and will convince the RL community that causality is an adequate tool to reason about observational data, which is plentiful in the world.

ETHICS STATEMENT

Confounding is a prevalent issue in human-generated data, and can be an important source of bias in the design of decision policies, if not dealt with properly. This paper, although theoretical, makes a humble step towards more robustness and fairness in AI-based decision systems, by combining causality and statistical learning to address the confounding problem. As such this work has potentially important societal implications, in particular in critical systems where lives are at stake such as medicine or self-driving cars, where human-generated data is prevalent.

REPRODUCIBILITY STATEMENT

Our notations, our POMDP and our causal frameworks are explicitly introduced in Sections 2 and 3. Our problem statement is clearly laid down in Section 4 before we present our contribution contribution, and the proofs of all our theoretical results are presented in the appendix, Section A.7. Our experimental setting is described briefly in the main body of the paper, Section 6, and in details in the appendix, Section A.5. The experimental results presented in the paper are reproducible, as both the workflow (scripts, parameters and seeds), and the source code are made publicly available alongside the paper.

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

# A  APPENDIX

## A.1  DO-CALCULUS

Several frameworks exist in the literature for reasoning about causality [23; 12]. Here we follow the framework of Judea Pearl, whose concept of *ladder of causation* is particularly relevant to answer RL questions. The first level of the ladder, *association*, relates to the observation of an external agent acting in the environment, while the second level, *intervention*, relates the question of what will happen to the environment as a result of one's own actions. The tool of do-calculus [25] acts as a bridge between these two levels, and relates interventional distributions, such as $p(y|do(x))$, to observational distributions, such as $p(y|x)$, in causal systems that can be expressed as DAGs. In a nutshell, do-calculus allows for measuring changes in the distribution of random variables $\{X, Y, Z, \dots\}$, when one performs an arbitrary intervention $do(x)$ which forces some variables to take values $X = x$ regardless of their causal ancestors. It relies on a complete set of rules [11; 27], which allow for the following equivalences when specific structural conditions are met in the causal DAG:

- R1: insertion/deletion of observations $p(y|do(x), z, w) = p(y|do(x), w)$,

- R2: action/observation exchange $p(y|do(x), do(z), w) = p(y|do(x), z, w)$,

- R3: insertion/deletion of actions $p(y|do(x), do(z), w) = p(y|do(x), w)$.

We refer the reader to Pearl [25] for a thorough introduction to $do$-calculus.

## A.2  ABOUT IGNORABILITY AND EXOGENEITY

In this paper we discuss and use at great length the concept of confounding, which is a core idea in Judea Pearl's do-calculus framework. For readers who are more familiar with the framework of potential outcomes from Donald Rubin [12], the concept of confounding closely relates to the concepts of ignorability and exogeneity. Indeed, both those concepts are shown to be equivalent to the unconfoundedness (no confounding) assumption in [23].

## A.3  GUIDING EXAMPLE: THE DOOR PROBLEM

This section introduces the *door* problem, and constitutes a guiding example that is meant to accompany the paper.

**The door problem**  Consider a door, a light, and two buttons A and B. The light is red 60% of the time, and green the rest of the time. When the light is red, button A opens the door, while when the light is green, then button B opens the door. I am told that the mechanism responsible for opening the door depends on both the light color and the button pressed (*light → door ← button*), but I am not given the mechanism itself. Suppose now that I am colorblind, and I want to open the door. Which button should I press ? In the do-calculus framework, the question I am asking is

$$\arg\max_{button \in \{A, B\}} p(door{=}open|do(button)).$$

**Interventional regime**  If I am able to observe myself or another colorblind person interacting with the door, then I know that which button is pressed is unrelated to which color the light is (*light ↛ button*). Then I can directly estimate the causal effect of the button on the door,

$$p(door{=}open|do(button)) = p(door{=}open|button).$$

Whichever policy is used to collect (*button*, *door*) samples[8], eventually I realise that button A has more chances of opening the door (60%) than button B (40%), and thus is the optimal choice.

---

[8]One assumption though is strict positivity, $\pi(button) > 0 \; \forall button$, so that both buttons are pressed.

**Observational regime**  Assume now that I observe another person interacting with the door. I do not know whether that person is colorblind or not (*light* → *button* is possible). Then, without further knowledge, I cannot recover the causal queries $p(door{=}open|do(button))$ from the observed distribution $p(door, button)$. In the $do$-calculus framework, the queries are said *non identifiable*. However, if that person was to tell me the light color they see before they press A or B, then I could recover those queries as follows,

$$p(door{=}open|do(button)) = \sum_{light \in \{red, green\}} p(light)p(door{=}open|light, button).$$

This formula, called *deconfounding*, eventually yields the correct causal transition probabilities regardless of the observed policy[9], given that enough (*light, button, door*) samples are observed.

**Merging interventional and observational data**  Let us now look at our door example in light of Theorem 1. Assume this time that I observe many (*button, door*) interactions from a non-colorblind person ($i = 0$), who's policy is $\pi(button{=}A|light{=}red) = 0.9$ and $\pi(button{=}A|light{=}green) = 0.4$. Then I can already infer from Theorem 1 that $p(door{=}open|do(button{=}A)) \in [0.54, 0.84]$ and $p(door{=}open|do(button{=}B)) \in [0.24, 0.94]$. I now get a chance to interact with the door ($i = 1$), and I decide to press $A$ 10 times and $B$ 10 times. I am unlucky, and my interventional study results in the following probabilities: $q(door{=}open|do(button{=}A)) = 0.5$ and $q(door{=}open|do(button{=}B)) = 0.5$. This does not coincide with my (reliable) observational study, and therefore I adjust $q(door{=}open|do(button{=}A))$ to its lower bound $0.54$. I now believe that pressing $A$ is more likely to be my optimal strategy.

### A.4  Recovering the standard POMDP transition model.

Recovering $\hat{q}(o_{t+1}|h_t, a_t, i = 1)$ can be done as follows:

$$\hat{q}(o_{t+1}|h_t, a_t, i = 1) = \sum_{z_t}^{\mathcal{Z}} \hat{q}(z_t|h_t, i = 1) \sum_{z_{t+1}}^{\mathcal{Z}} \hat{q}(z_{t+1}|z_t, a_t)\hat{q}(o_{t+1}|z_{t+1}).$$

The second and third terms are readily available as components of the augmented POMDP model $\hat{q}$, while the first term can be recovered by unrolling a forward algorithm over the augmented DAG structure. First, we have

$$\hat{q}(z_0, o_0|i = 1) = \hat{q}(z_0)\hat{q}(o_0|z_0),$$

$$\hat{q}(z_0|h_0, i = 1) = \frac{\hat{q}(z_0, o_0|i = 1)}{\sum_{z_0}^{\mathcal{Z}} \hat{q}(z_0, o_0|i = 1)}.$$

Then, for every $t'$ from 0 to $t - 1$,

$$\hat{q}(z_{t'+1}, o_{t'+1}|h_{t'}, a_{t'}, i = 1) = \sum_{z_{t'}}^{\mathcal{Z}} \hat{q}(z_{t'}|h_{t'}, i = 1)\hat{q}(z_{t'+1}|z_{t'}, a_{t'})\hat{q}(o_{t'+1}|z_{t'+1}),$$

$$\hat{q}(z_{t'+1}|h_{t'+1}, i = 1) = \frac{\hat{q}(z_{t'+1}, o_{t'+1}|h_{t'}, a_{t'}, i = 1)}{\sum_{z_{t'+1}}^{S} \hat{q}(z_{t'+1}, o_{t'+1}|h_{t'}, a_{t'}, i = 1)}.$$

---

[9]The strict positivity condition here is $\pi(button|light) > 0 \ \forall button, light$.

A.5 EXPERIMENTAL DETAILS

The code for reproducing our experiments is made available online[10].

We perform experiments on three synthetic toy problems: the *door* problem described earlier (Section A.3), the classical *tiger* problem from the literature [4], and a 5x5 *gridworld* problem inspired from Alt et al. [1].

**Data** To assess the performance of our method, we consider a large observational dataset $\mathcal{D}_{obs}$ of fixed size (512 samples for *door*, 8192 samples for *tiger* and *gridworld*), and an interventional dataset $\mathcal{D}_{int}$ of varying size, ranging on an exponential scale from 4 to $|\mathcal{D}_{obs}|$.

**Baselines** We compare the performance of the transition model $\hat{q}$ recovered in three different settings: *no obs*, when only interventional data ($\mathcal{D} = \mathcal{D}_{int}$) is used for training; *naive*, when observational data is naively combined with interventional data as if there was no confounding ($\mathcal{D} = \mathcal{D}_{int} \cup \{(\tau, 1)|(\tau, i) \in \mathcal{D}_{obs})\}$); and *augmented*, our proposed method ($\mathcal{D} = \mathcal{D}_{int} \cup \mathcal{D}_{obs}$). Note that the only difference between each of those settings is the training dataset, all other aspects (learning procedure, model architecture, loss function) begin the same.

**Training** In all our experiments we use a tabular model for $\hat{q}$, that is, we use discrete probability tables for each building blocs of the transition model, $q(z_0)$, $q(o_t|z_t)$, $q(z_{t+1}|z_t, a_t)$, and $q(a_t|h_t, z_t, i = 0)$. We use a latent space $|\mathcal{Z}|$ of size 32, 32 and 128 respectively for each toy problem, while the true latent space $|\mathcal{S}|$ is of size 3, 6 and 42. We train $\hat{q}$ by directly minimizing the negative log likelihood (4) via gradient descent. We use the Adam optimizer [14] with a learning rate of $10^{-2}$, and train for 500 epochs consisting of 50 gradient descent steps with minibatches of size 32. We divide the learning rate by 10 after 10 epochs without loss improvement (reduce on plateau), and we stop training after 20 epochs without improvement (early stopping). In the *door* experiment we derive the optimal policy $\hat{\pi}^\star$ exactly, while in the *tiger* and *gridworld* experiments we train a "dreamer" RL agent on imaginary samples $\tau \sim \hat{q}(\tau|i = 1)$ obtained from the model, using the belief states $\hat{q}(s_t|h_t)$ as features. We use a simple Actor-Critic algorithm for training, and our agents consist of a simple MLP with one hidden layers for both the critic and the policy parts. RL agents are trained until convergence or with a maximum number of 1000 epochs, with a learning rate of $10^{-2}$, a discount factor $\gamma = 0.9$ and a batch size of 8.

**JS divergence** To evaluate the general quality of the recovered transition models, we compute the expected Jensen-Shannon divergence between the learned $\hat{q}(o_{t+1}|h_t, i = 1)$ and the true $p(o_{t+1}|h_t, i = 1)$, over transitions generated using a uniformly random policy $\pi_{rand}$,

$$\frac{1}{2}\mathbb{E}_{\tau \sim p_{init}, p_{trans}, p_{obs}, \pi_{rand}} \left[ \log \frac{p(o_0)}{m(o_0)} + \sum_{t=1}^{|\tau|} \log \frac{p(o_{t+1}|h_t, i = 1)}{m(o_{t+1}|h_t, i = 1)} \right]$$

$$+ \frac{1}{2}\mathbb{E}_{\tau \sim \hat{q}_{init}, \hat{q}_{trans}, \hat{q}_{obs}, \pi_{rand}} \left[ \log \frac{\hat{q}(o_0)}{m(o_0)} + \sum_{t=1}^{|\tau|} \log \frac{\hat{q}(o_{t+1}|h_t, i = 1)}{m(o_{t+1}|h_t, i = 1)} \right],$$

where $m(.) = \frac{1}{2}(\hat{q}(.) + p(.))$. In the first experiment we compute the JS exactly, while in the second experiment we use a stochastic approximation over 100 trajectories $\tau$ to estimate each of the expectation terms in the JS empirically.

**Reward.** To evaluate quality of the recovered transition models for solving the original RL task, that is, maximizing the expected long-term reward, we evaluate the policy $\hat{\pi}^\star$, obtained after planning with the recovered model $\hat{q}$, on the true environment $p$,

$$\mathbb{E}_{\tau \sim p_{init}, p_{trans}, p_{obs}, \hat{\pi}^\star} \left[ \sum_{t=0}^{|\tau|} R(o_t) \right].$$

In the first experiment we compute this expectation exactly, while in the second experiment we use a stochastic approximation using 100 trajectories $\tau$.

---

[10]https://supplementary.materials/disclosed.after.acceptance

## A.6 Complete empirical results

### A.6.1 Door experiment

The *door* experiment corresponds to a simple binary bandit setting, that is, a specific POMDP with horizon $H = 1$. The observation space is of size $|\mathcal{O}| = 0$, since the learning agent receives no observation, and the hidden state space is of minimal size $|\mathcal{S}| = 3$ to encode both the initial light color and the reward obtained afterwards. The bandit dynamics are described in Table 1.

| *light* | |
|---|---|
| red | green |
| 0.6 | 0.4 |

$p(\textit{light})$

| | | door | |
|---|---|---|---|
| *light* | *button* | closed | open |
| red | A | 0.0 | 1.0 |
| red | B | 1.0 | 0.0 |
| green | A | 1.0 | 0.0 |
| green | B | 0.0 | 1.0 |

$p(\textit{door}|\textit{light},\textit{button})$

Table 1: Probability tables for our *door* bandit problem.

We repeat the *door* experiment in six different scenarios, corresponding to different privileged policies $\pi_{prv}$ ranging from a totally random agent to a perfectly good and a perfectly bad agent. Each time, we evaluate the performance of the *no obs*, *naive* and *augmented* approaches under different data regimes, by varying the sample size for both the observational data $\mathcal{D}_{obs}$ and the interventional data $\mathcal{D}_{int}$ in the range $(4, 8, 16, 32, 64, 128, 256, 512)$.

In each scenario, we report both the expected reward and the JS as heatmaps with $|\mathcal{D}_{int}|$ and $|\mathcal{D}_{obs}|$ in the $x$-axis and $y$-axis respectively, to highlight the combined effect of the sample sizes on each approach. We also provide as a heatmap the difference between our approach, *augmented*, and the two other approaches *no obs* and *naive*. We always plot the expected reward in the first row, and JS in the second row. As a remark, shades of green show gains in reward (the higher the better), while shades of purple show gains in JS (the lower the better).

Finally, we also present two plots which provide a focus on the data regime that corresponds to the largest number of observational data ($|\mathcal{D}_{obs}| = 512$), as in the main paper.

**Noisy Good Expert**    In the noisy good expert setting, the expert plays halfway between a perfect and a random policy. The diversity of its action leads to a good start for the *naive* model but the bias it contains is hard to overcome. In contrast, our method makes good use of the observational data from the start and is also able to correct the bias as interventional data come in, eventually converging towards the true transition model.

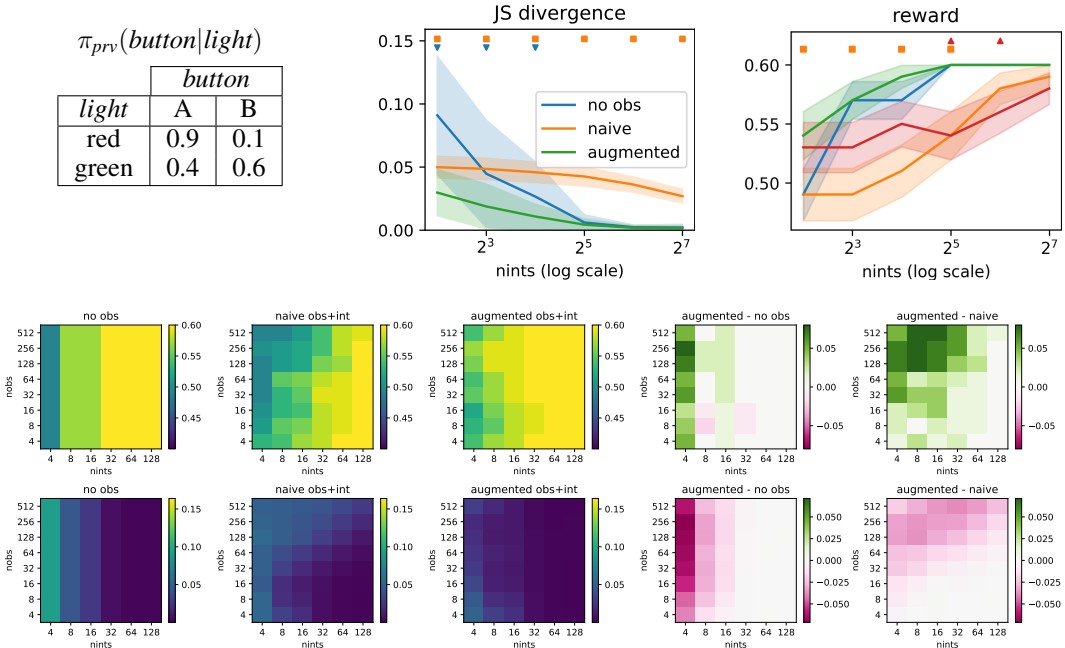

Figure 7: Noisy good expert setting. Heatmaps correspond respectively to the expected reward (top row, higher is better) and the JS divergence (bottom row, lower is better).

**Random Expert**  A random policy naturally results in unconfounded observational data, since it does not exploits the privileged information. Hence, the *naive* approach is unbiased in this case, and actually makes the best use of the observational data. Our approach, *augmented*, exhibits an overall comparable performance, only slightly worse at times. We believe this can be explained by the additional complexity of our method which tries to disentangle a confounded regime in the data, and is not best suited to unconfounded data.

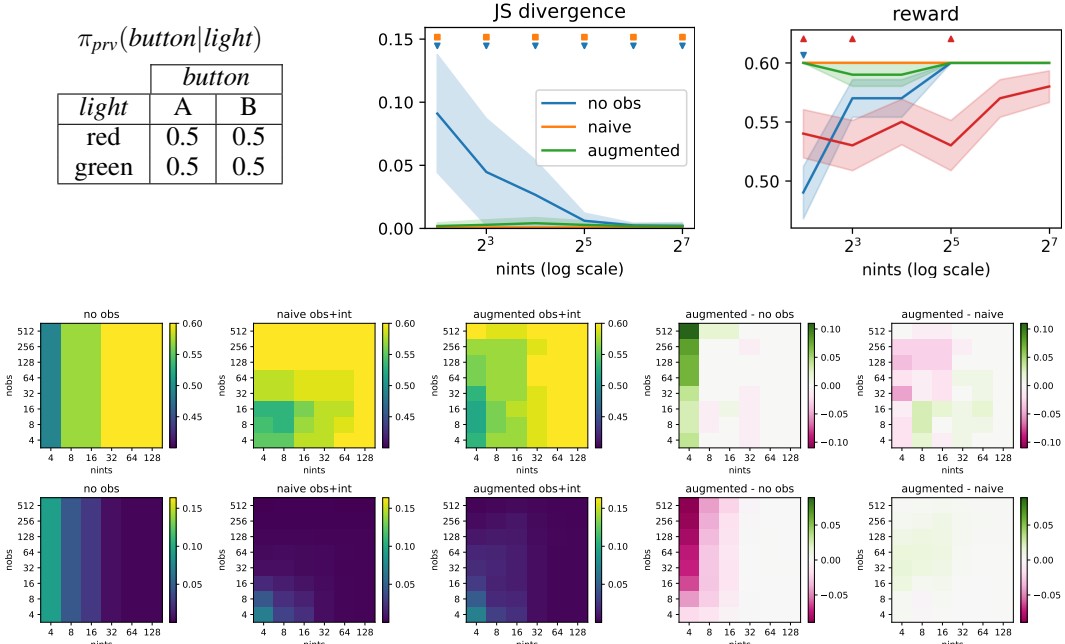

Figure 8: Random expert setting. Heatmaps correspond respectively to the expected reward (top row, higher is better) and the JS divergence (bottom row, lower is better).

**Perfectly Good Expert**    Observing a perfectly good expert playing in the *door* problem induces a strong bias, because every observed action always results in a positive reward. As such, the *naive* approach struggles to learn a good transition model. The bias however is quickly corrected by our *augmented* approach, which eventually converges to the true transition model faster than the *no obs* approach.

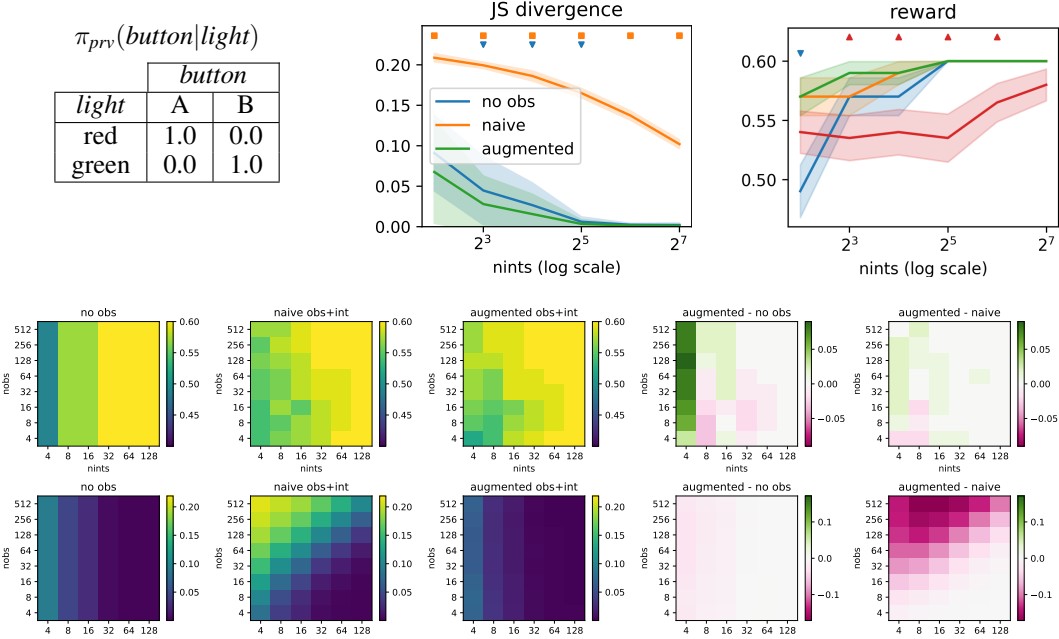

Figure 9: Perfectly good expert setting. Heatmaps correspond respectively to the expected reward (top row, higher is better) and the JS divergence (bottom row, lower is better).

**Perfectly Bad Expert** Similarly to the previous setting, observing an expert that always chooses a bad action leads to a strong bias, as every action is associated to a low reward. The behaviour in terms of JS and reward is similar as well.

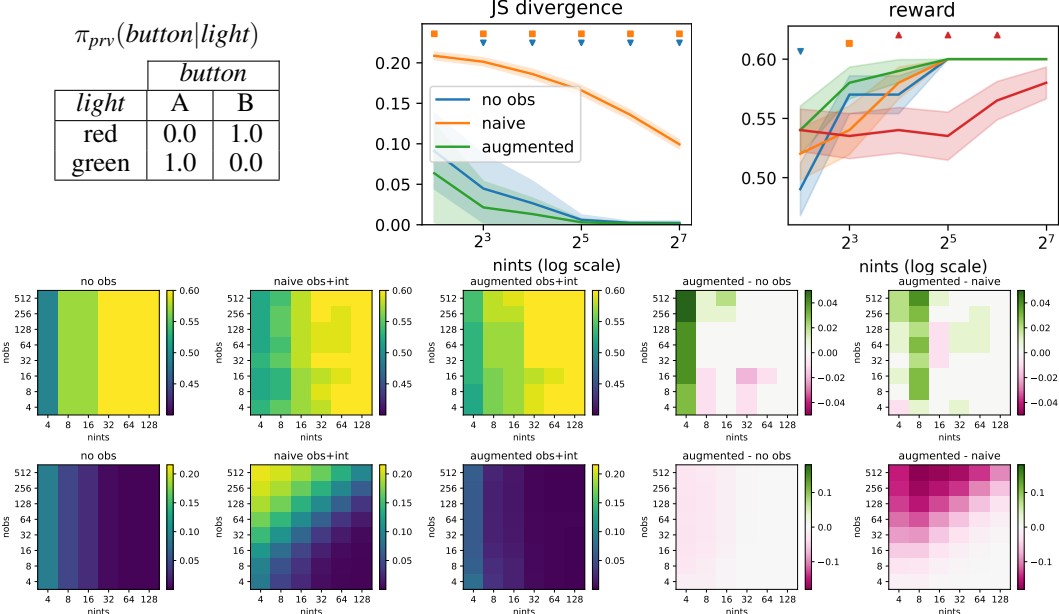

Figure 10: Perfectly bad expert setting. Heatmaps correspond respectively to the expected reward (top row, higher is better) and the JS divergence (bottom row, lower is better).

**Positively Biased Expert** Here the expert's policy is considered as *positively biased* in the sense that the agent will only obtain a positive reward when playing button A (with 55% chances) and never by playing button B (0% chances). Because playing button A is actually the optimal policy, this strong bias has a positive effect on the reward for the *naive* approach. Hence, although worse in terms of JS than our approach, the *naive* approach always results in a very good policy in terms of reward. Our *augmented* approach, however, seems more conservative.

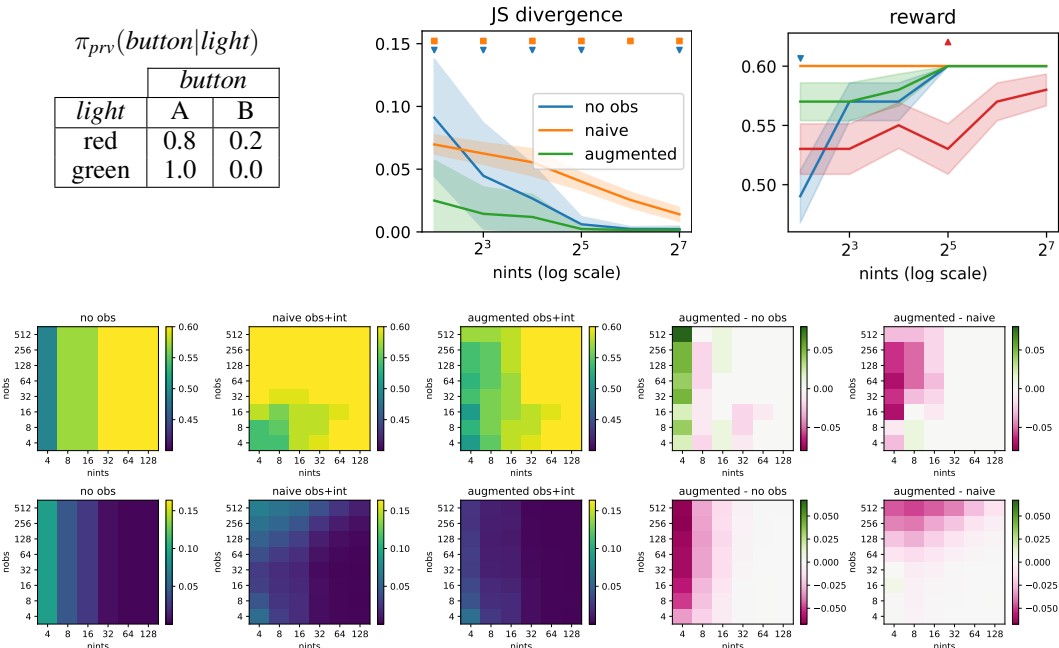

Figure 11: Positively biased expert setting. Heatmaps correspond respectively to the expected reward (top row, higher is better) and the JS divergence (bottom row, lower is better).

**Negatively Biased Expert**   In an analogous way, a negatively biased expert will overuse button A, leading to mixed feelings regarding this button, whereas it will always get a positive reward each time it uses button B. This leads to the opposite behavior as we had in the previous setting, with the *naive* approach always favoring the use of button B, and obtaining a bad performance in terms of reward. The *naive* approach only gets better when a lot of interventional data is combined with the biased observational data, while our *augmented* approach is able to overcome this pessimistic bias very early on, and converges faster than both *no obs* and *naive*.

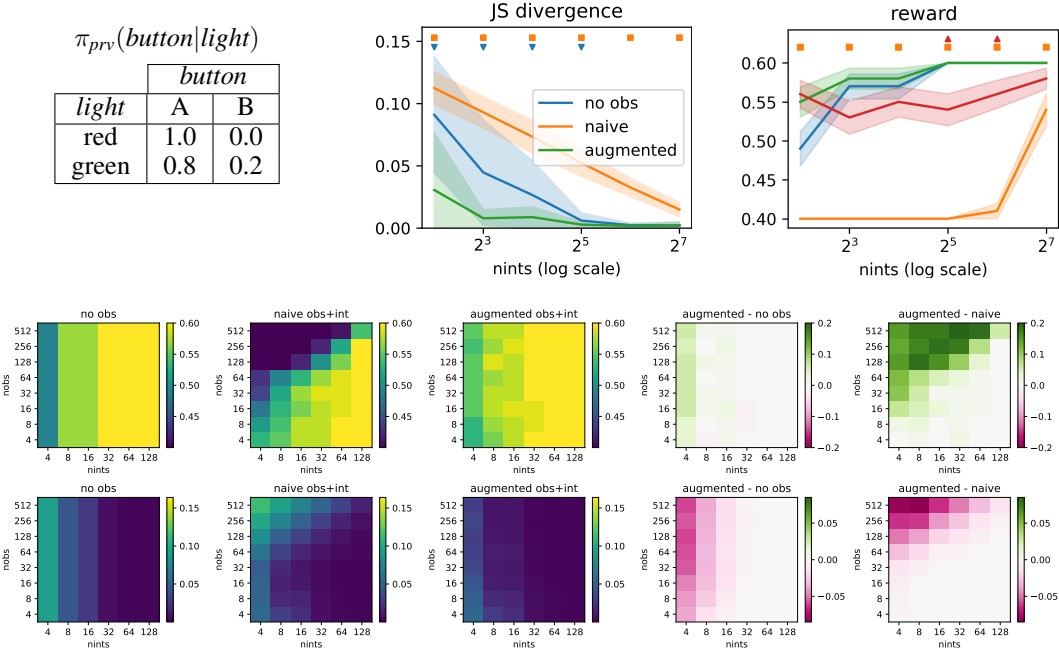

Figure 12: Pessimistic bias expert setting. Heatmaps correspond respectively to the expected reward (top row, higher is better) and the JS divergence (bottom row, lower is better).

A.6.2 TIGER EXPERIMENT

The *tiger* experiment corresponds a synthetic POMDP toy problem proposed by Cassandra et al. [4]. In short, in this problem the agent stands in front of two doors to open, one of them having a tiger behind it (-100 reward), and the other one a treasure (+10 reward). The agent also gets a noisy observation of the system in the form of the roar from the tiger, which seems to originate from the correct door most of the time (85% chances) and the wrong door sometimes (15% chances). In order to reduce uncertainty the agent can listen to the tiger's roar again, at the cost of a small penalty (-1). We present the simplified POMDP dynamics in Table 2, and in our experiments we impose a fixed horizon of size $H = 50$. The observation space is of size $|\mathcal{O}| = 6$, to encode the roar location perceived by the agent and the obtained reward, $o_t = (roar_t, reward_t)$, and the hidden state space is of minimal size $|\mathcal{S}| = 6$ to encode both the tiger position and the reward obtained at each time step, $s_t = (tiger_t, reward_t)$.

| $tiger_0$ | |
|---|---|
| left | right |
| 0.5 | 0.5 |

$p(tiger_0)$

| $tiger_t$ | $roar_t$ | |
|---|---|---|
| | left | right |
| left | 0.85 | 0.15 |
| right | 0.15 | 0.85 |

$p(roar_t|tiger_t)$

| $tiger_t$ | $action_t$ | $tiger_{t+1}$ | |
|---|---|---|---|
| | | left | right |
| left | listen | 1.0 | 0.0 |
| | open left | 0.5 | 0.5 |
| | open right | 0.5 | 0.5 |
| right | listen | 0.0 | 1.0 |
| | open left | 0.5 | 0.5 |
| | open right | 0.5 | 0.5 |

$p(tiger_{t+1}|tiger_t, action_t)$

| $tiger_t$ | $action_t$ | $reward_{t+1}$ | | |
|---|---|---|---|---|
| | | -1 | -100 | +10 |
| left | listen | 1.0 | 0.0 | 0.0 |
| | open left | 0.0 | 1.0 | 0.0 |
| | open right | 0.0 | 0.0 | 1.0 |
| right | listen | 1.0 | 0.0 | 0.0 |
| | open left | 0.0 | 0.0 | 1.0 |
| | open right | 0.0 | 1.0 | 0.0 |

$p(reward_{t+1}|tiger_t, action_t)$

Table 2: Probability tables for the *tiger* problem.

For the tiger experiment we consider four different privileged policies $\pi_{prv}$ for the observed agent. We then evaluate the performance of the *no obs*, *naive* and *augmented* approaches under different data regimes, by keeping the observational data fixed to $|\mathcal{D}_{obs}| = 8192$ while varying the varying the number of interventional data for $\mathcal{D}_{int}$ in the range $(4, 8, 16, 32, 64, 128, 256, 512, 1024, 2048, 4096, 8192)$.

**Noisy Good Expert**   In this scenario the privileged expert adopts a policy that plays the optimal action most of the time (open the treasure door), but also sometimes decides to just listen or to open the wrong door. As can be seen, in this scenario our *augmented* method makes the best use of the observational data, and is significantly better than both the *no obs* and *naive* approaches in the low-sample regime, both in terms of quality of the estimated transition model and obtained reward.

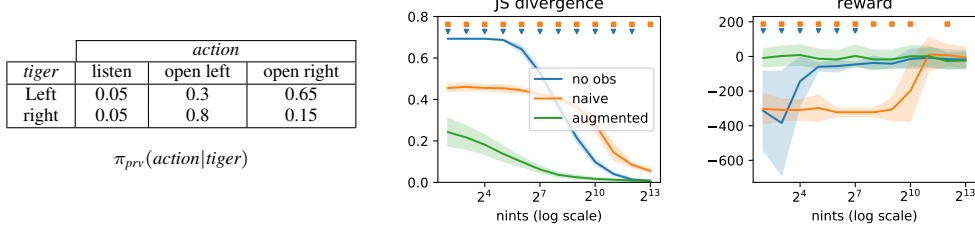

|  | *action* | | |
|---|---|---|---|
| *tiger* | listen | open left | open right |
| Left | 0.05 | 0.3 | 0.65 |
| right | 0.05 | 0.8 | 0.15 |

$\pi_{prv}(action|tiger)$

Figure 13: Noisy good agent.

**Random Expert**   In the random scenario there is no confounding, and observational data can be safely mixed with interventional data. The *naive* approach thus does not suffer from any bias, and in fact is the one that converges the fastest to the optimal transition model and policy. Our method, while it manages to leverage the observational data to converge faster than *no obs*, suffers from a worse performance than *naive* in the low sample regime, most likely because it tries to recover a spurious confounding variable to distinguish the observational and interventional regimes, when none actually exists.

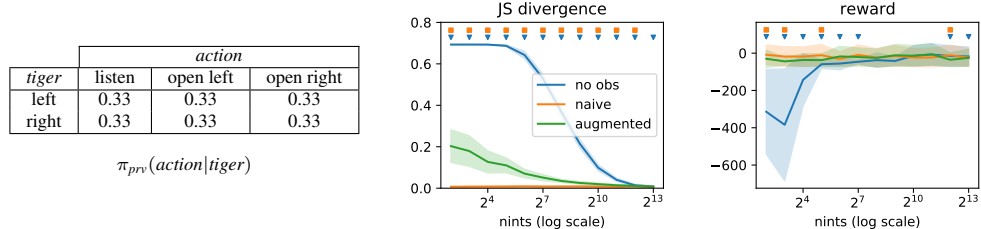

|  | *action* | | |
|---|---|---|---|
| *tiger* | listen | open left | open right |
| left | 0.33 | 0.33 | 0.33 |
| right | 0.33 | 0.33 | 0.33 |

$\pi_{prv}(action|tiger)$

Figure 14: Random agent.

**Very Good Expert**   Here the privileged expert never opens the wrong door, and thus never receives the very penalizing -100 reward. As a result the *naive* approach seems to be overly optimistic with respect to the action of opening a door, which strongly affects the expected reward it obtains in the true environment. While our *augmented* approach seems also to suffer from this bias in the very low sample regime (as can be seen on the reward plot), overall the quality of the recovered transition model is still superior to both other approaches, and converges faster to the true transition model.

| | action | | |
|---|---|---|---|
| tiger | listen | open left | open right |
| left | 0.05 | 0.0 | 0.95 |
| right | 0.05 | 0.95 | 0.0 |

$\pi_{prv}(action|tiger)$

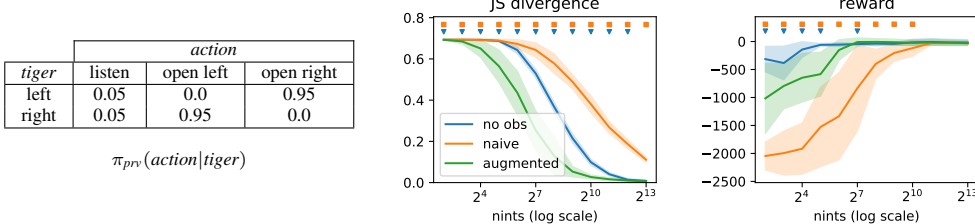

Figure 15: Very good agent.

**Very Bad Expert**   Here the privileged expert never opens the correct door, and thus never receives a positive reward (+10). As a result, the *naive* approach seems to be very conservative, and prefers not to take any chances opening a door. It turns out that this strategy is not too bad in terms of reward (always listening yields a -51 total reward), and as such this causal bias seems to positively affect the performance of the *naive* approach in the low sample regime, but prevents it from obtaining a better policy in the high sample regime too. Our *augmented* method, on the other hand, is able to escape this overly conservative strategy earlier on, and converges to a good-performing policy faster than both other approaches.

| | action | | |
|---|---|---|---|
| tiger | listen | open left | open right |
| left | 0.05 | 0.95 | 0.0 |
| right | 0.05 | 0.0 | 0.95 |

$\pi_{prv}(action|tiger)$

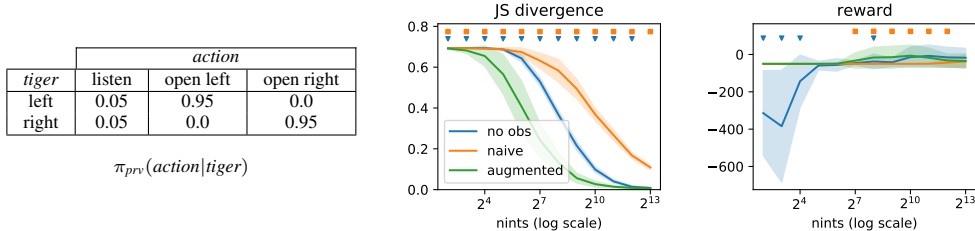

Figure 16: Very bad agent.

### A.6.3 GRIDWORLD EXPERIMENT

The *gridworld* experiment, represented in Figure 17, is inspired from [1]. It consists in a small 5x5 grid where the agent starts on the top-left corner, and tries to get to a target placed on the bottom side behind a large wall. The agent can use five actions: *top*, *right*, *bottom*, *left* and *idle*, and only receives a noisy signal about its current position. At each time step, the agent's position is revealed with 20% chances, and remains completely hidden otherwise. In addition, the agent's actions only have a stochastic effect, i.e., the agent moves into the desired direction with 50% chances, and otherwise slips at random to one of the 5 adjacent tiles or current tile. In case the agent would bump into a wall, it simply remains at its current position. The observation space is of size $|\mathcal{O}| = 44$, to encode both the agent's location (or the indication that the location is hidden) and the reward, and the hidden state space is of size $|\mathcal{S}| = 42$ to encode both the agent's location and the reward. In this experiment we impose a fixed horizon of size $H = 20$.

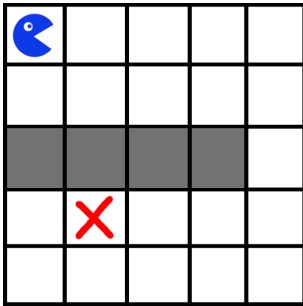

Figure 17: The gridworld problem

For the gridworld experiment we consider a single policy $\pi_{prv}$ for the privileged agent, who acts optimally (shortest path from current location to target). We then evaluate the performance of the *no obs*, *naive* and *augmented* approaches under different data regimes, by keeping the observational data fixed to $|\mathcal{D}_{obs}| = 8192$ while varying the varying the number of interventional data for $\mathcal{D}_{int}$ in the range $(4, 8, 16, 32, 64, 128, 256, 512, 1024, 2048, 4096, 8192)$.

**Very Good Expert**   In this scenario the privileged agent adopts a perfect policy, and always chooses an action leading to the shortest path towards the target. As can be seen, here again our *augmented* method makes the best use of the observational data, and converges faster than both the *no obs* and the *naive* approaches for recovering the true transition model. This improvement in the transition model also translates into an improvement in terms of the learned policy, which starts converging towards high reward values with fewer samples ($2^7$) than both *no obs* and *naive* ($2^9$).

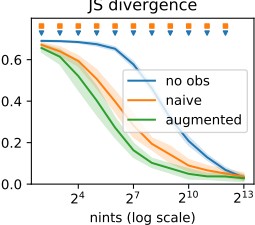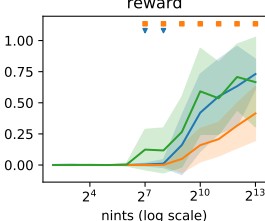

Figure 18: Perfect agent.

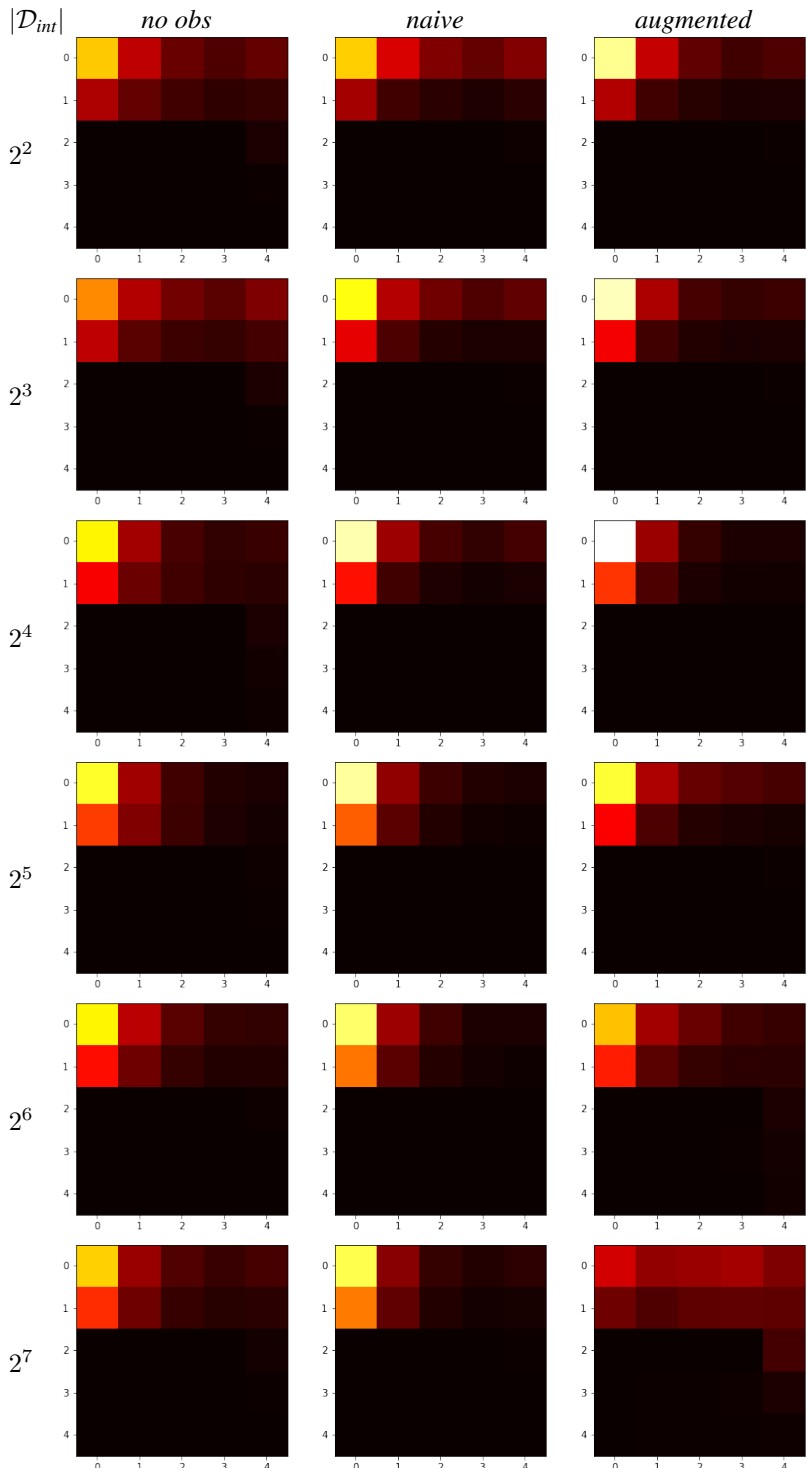

Figure 19: Average heat-maps over 100 episodes $\times$ 10 seeds, of the tiles visited by each trained agent (*no obs*, *naive*, *augmented*) for different interventional data sizes ($2^2$, $2^3$, $2^4$, $2^5$, $2^6$, $2^7$). The *augmented* approach is the fastest (in terms of interventional data) to learn how to properly escape the top part of the maze through tile $(4, 2)$, and then move towards the treasure on tile $(1, 3)$.

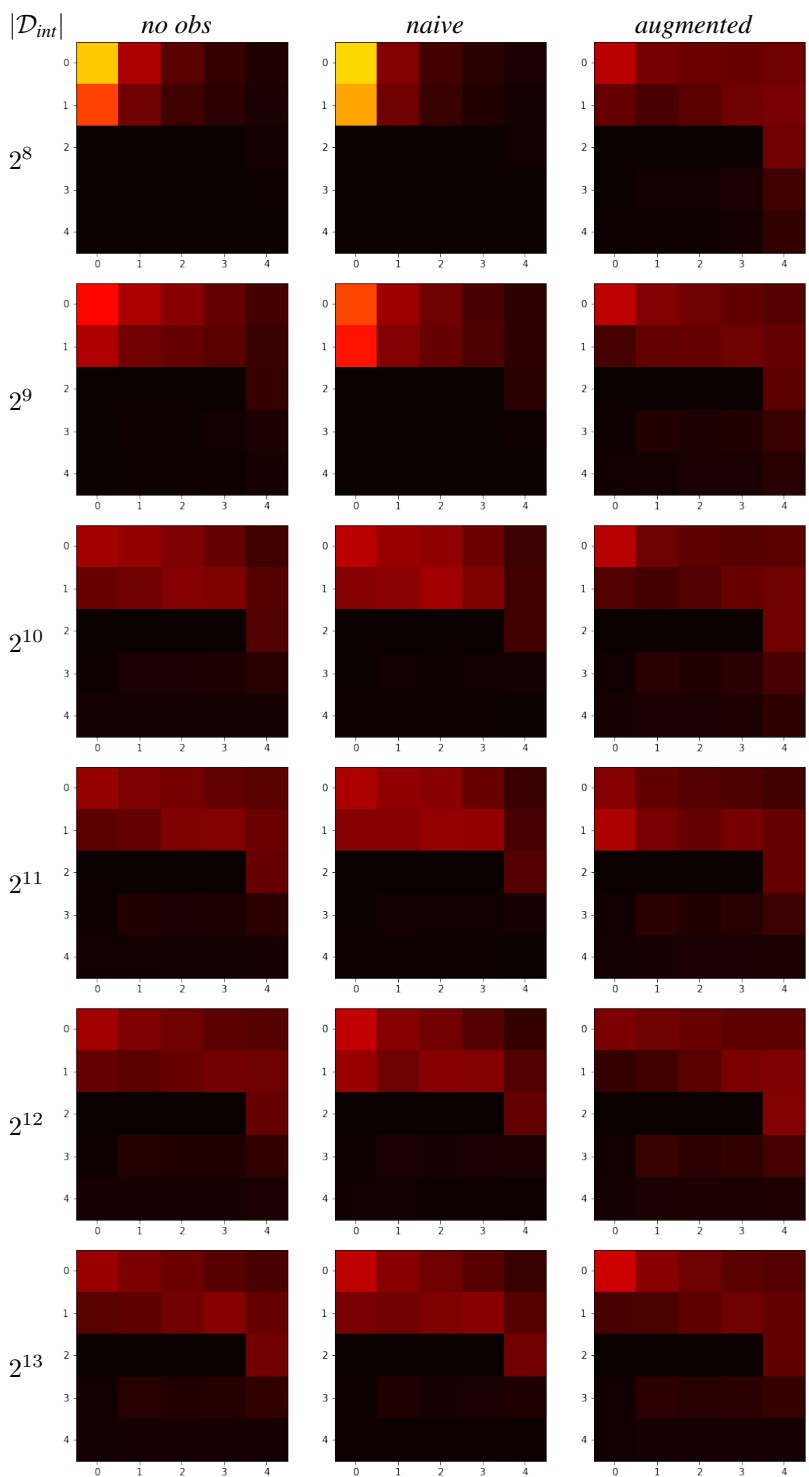

Figure 20: Average heat-maps over 100 episodes × 10 seeds, of the tiles visited by each trained agent (*no obs*, *naive*, *augmented*) for different interventional data sizes ($2^8$, $2^9$, $2^{10}$, $2^{11}$, $2^{12}$, $2^{13}$). The *augmented* approach is the fastest (in terms of interventional data) to learn how to properly escape the top part of the maze through tile (4, 2), and then move towards the treasure on tile (1, 3).

A.7   PROOFS.

**Proposition 1.** *Assuming $|\mathcal{Z}| \geq |\mathcal{S}|$, $\hat{q}(o_{t+1}|h_t, a_t, i = 1)$ is an unbiased estimator of $p(o_{t+1}|h_t, a_t, i = 1)$.*

*Proof.* The proof is straightforward. First, we have that $\mathcal{D} \sim p(\tau, i)$. Second, we have $p \in \mathcal{Q}$, because $\mathcal{Q}$ is only restricted to the augmented POMDP constraints, and because its latent space is sufficiently large ($|\mathcal{Z}| \geq |\mathcal{S}|$). Therefore, $\hat{q}(\tau, i)$ solution of (4) is an unbiased estimator of $p(\tau, i)$, and in particular $\hat{q}(o_{t+1}|h_t, a_t, i = 1)$ is an unbiased estimator of $p(o_{t+1}|h_t, a_t, i = 1)$. □

**Corollary 1.** *The estimator $\hat{q}(o_{t+1}|h_t, a_t, i = 1)$ recovered after solving (4) with $|\mathcal{D}_{obs}| \to \infty$ offers strictly better generalization guarantees than the one with $|\mathcal{D}_{obs}| = 0$, for any $\mathcal{D}_{int}$.*

*Proof.* There exists at least one history-action couple $(h_{T-1}, a_{T-1})$, $T \geq 1$, that has non-zero probability in the observational regime. This ensures that there exists a value $o_T$ for which $\prod_{t=0}^{T-1} p(a_t|h_t, i = 0)p(o_{t+1}|h_t, a_t, i = 0)$ is strictly positive, which in turn ensures $\hat{q}(o_{T+1}|h_T, a_T, i = 1) > 0$. As a result, the family of models $\{q(o_{t+1}|h_t, a_t, i = 1) \mid q \in \mathcal{Q}, q(\tau|i = 0) = p(\tau|i = 0)\}$ is a strict subset of the unrestricted family $\{q(o_{t+1}|h_t, a_t, i = 1) \mid q \in \mathcal{Q}\}$, and thus offers strictly better generalization guarantees. □

**Theorem 1.** *Assuming $|\mathcal{D}_{obs}| \to \infty$, for any $\mathcal{D}_{int}$ the recovered causal model is bounded as follows:*

$$\prod_{t=0}^{T-1} \hat{q}(o_{t+1}|h_t, a_t, i = 1) \geq \prod_{t=0}^{T-1} p(a_t|h_t, i = 0)p(o_{t+1}|h_t, a_t, i = 0), \text{ and}$$

$$\prod_{t=0}^{T-1} \hat{q}(o_{t+1}|h_t, a_t, i = 1) \leq \prod_{t=0}^{T-1} p(a_t|h_t, i = 0)p(o_{t+1}|h_t, a_t, i = 0) + 1 - \prod_{t=0}^{T-1} p(a_t|h_t, i = 0),$$

$\forall h_{T-1}, a_{T-1}, T \geq 1$ *where* $p(h_{T-1}, a_{T-1}, i = 0) > 0$.

*Proof of Theorem 1.* Consider $q(\tau, i) \in \mathcal{Q}$ any distribution that follows our augmented POMDP constraints. Then, for every $T \geq 1$ we have

$$\prod_{t=0}^{T-1} q(a_t|h_t, i)q(o_{t+1}|h_t, a_t, i) = \frac{q(\tau|i)}{q(h_0|i)}$$

$$= \sum_{z_{0 \to T}}^{\mathcal{Z}^{T+1}} q(z_0|h_0, i) \prod_{t=0}^{T-1} q(a_t, z_{t+1}, o_{t+1}|z_t, h_t, i),$$

by using $A_t, Z_{t+1}, O_{t+1} \perp\!\!\!\perp Z_{0 \to t-1} \mid Z_t, H_t, I$, which can be read via $d$-separation in the augmented POMDP DAG. Likewise, for every $t \geq 0$ we have

$$q(o_{t+1}|h_t, a_t, i = 1) = \sum_{z_{t+1}}^{\mathcal{Z}} q(z_{t+1}, o_{t+1}|h_t, a_t, i = 1)$$

$$= \sum_{z_t}^{\mathcal{Z}} q(z_t|h_t, i = 1) \sum_{z_{t+1}}^{\mathcal{Z}} q(z_{t+1}, o_{t+1}|z_t, h_t, a_t, i = 0),$$

by using $Z_t \perp\!\!\!\perp A_t \mid H_t, I = 1$ and $Z_{t+1}, O_{t+1} \perp\!\!\!\perp I \mid Z_t, A_t, H_t$. Then for every $t \geq 1$ we can further write

$$q(o_{t+1}|h_t, a_t, i = 1) = \sum_{z_t}^{\mathcal{Z}} \frac{q(z_t, o_t|h_{t-1}, a_{t-1}, i = 1)}{q(o_t|h_{t-1}, a_{t-1}, i = 1)} \sum_{z_{t+1}}^{\mathcal{Z}} q(z_{t+1}, o_{t+1}|z_t, h_t, a_t, i = 0).$$

By recursively decomposing every $q(z_t, o_t|h_{t-1}, a_{t-1}, i = 1)$ until $t = 0$, and finally by using $Z_0 \perp\!\!\!\perp I \mid H_0$, we obtain that for any $T \geq 1$

$$\prod_{t=0}^{T-1} q(o_{t+1}|h_t, a_t, i = 1) = \sum_{z_{0 \to T}}^{\mathcal{Z}^{T+1}} q(z_0|h_0, i = 0) \prod_{t=0}^{T-1} q(z_{t+1}, o_{t+1}|z_t, a_t, h_t, i = 0),$$

which can be re-expressed as

$$\prod_{t=0}^{T-1} q(o_{t+1}|h_t, a_t, i = 1) = \sum_{a'_{0 \to T-1}}^{\mathcal{A}^T} \sum_{z_{0 \to T}}^{\mathcal{Z}^{T+1}} q(z_0|h_0, i = 0) \prod_{t=0}^{T-1} q(a'_t|z_t, h_t, i = 0) q(z_{t+1}, o_{t+1}|z_t, h_t, a_t, i = 0).$$

By considering the case $a'_{0 \to T-1} = a_{0 \to T-1}$ in isolation, and by assuming probabilities are positive, we readily obtain our first bound,

$$\prod_{t=0}^{T-1} q(o_{t+1}|h_t, a_t, i = 1) \geq \prod_{t=0}^{T-1} q(a_t|h_t, i = 0) q(o_{t+1}|h_t, a_t, i = 0).$$

In order to obtain our second bound, we further isolate the cases $a'_0 \neq a_0$, then $a'_0 = a_0 \wedge a'_1 \neq a_1$, then $a'_0 = a_0 \wedge a'_1 = a_1 \wedge a'_2 \neq a_2$ and so on until $a'_{0 \to T-2} = a_{0 \to T-2} \wedge a'_{T-1} \neq a_{T-1}$, which yields

$$\prod_{t=0}^{T-1} q(o_{t+1}|h_t, a_t, i = 1) = \prod_{t=0}^{T-1} q(a_t|h_t, i = 0) q(o_{t+1}|h_t, a_t, i = 0)$$

$$+ \sum_{z_{0 \to T}}^{\mathcal{Z}^{T+1}} q(z_0|h_0, i = 0) \left(1 - q(a_0|z_0, h_0, i = 0)\right) \prod_{t=0}^{T-1} q(z_{t+1}, o_{t+1}|z_t, h_t, a_t, i = 0)$$

$$+ \sum_{K=0}^{T-2} \sum_{z_{0 \to T}}^{\mathcal{Z}^{T+1}} q(z_0|h_0, i = 0) \prod_{t=0}^{K} q(a_t, z_{t+1}, o_{t+1}|z_t, h_t, i = 0) \left(1 - q(a_K|z_K, h_K, i = 0)\right)$$

$$\prod_{t=K+1}^{T-1} q(z_{t+1}, o_{t+1}|z_t, h_t, a_t, i = 0).$$

Then by assuming probabilities are upper bounded by 1, we obtain

$$\prod_{t=0}^{T-1} q(o_{t+1}|h_t, a_t, i = 1) \leq \prod_{t=0}^{T-1} q(a_t|h_t, i = 0) q(o_{t+1}|h_t, a_t, i = 0) + 1 - q(a_0|h_0, i = 0)$$

$$+ \sum_{K=0}^{T-2} \prod_{t=0}^{K} q(o_{t+1}|h_t, a_t, i = 0) \left( \prod_{t=0}^{K-1} q(a_t|h_t, i = 0) - \prod_{t=0}^{K} q(a_t|h_t, i = 0) \right)$$

$$\leq \prod_{t=0}^{T-1} q(a_t|h_t, i = 0) q(o_{t+1}|h_t, a_t, i = 0) + 1 - \prod_{t=0}^{T-1} q(a_t|h_t, i = 0).$$

Finally, with $\hat{q}$ solution of (4) and $|\mathcal{D}_{obs}| \to \infty$ we have that $D_{KL}(p(\tau|i = 0)\|\hat{q}(\tau|i = 0)) = 0$, and thus $\hat{q}(a_t|h_t, i = 0) = p(a_t|h_t, i = 0)$ and $\hat{q}(o_{t+1}|h_t, a_t, i = 0) = p(o_{t+1}|h_t, a_t, i = 0)$, which shows the desired result. $\square$

