# OpenReview forum: "Causal Reinforcement Learning using Observational and Interventional Data"
_ICLR.cc/2022/Conference — ICLR 2022 Submitted_

### Official Review · Reviewer_8zjw · 2021-11-02

**Correctness:** 3
**Technical Novelty And Significance:** 2
**Empirical Novelty And Significance:** 1
**Recommendation:** 6
**Confidence:** 3

**Main Review:**

Strengths

The paper proposes to learn a latent causal transition model explaining both the interventional and observational data and infer the standard POMDP transition model via deconfounding using the recovered latent variable.

The  paper shows that combing both intervention data and observation data can achieve better generalization guarantees in the asymptotic case.

Weaknesses
The setting considered is rather limiting because it assumes that there is no confounder in the model of the POMDP.

The writing is very poor. The main contribution, section 4.3 is not clearly explained. It is not clear how imposing an observational distribution q(τ|i = 0) acts as a regularizer for the interventional distribution.

Experiments are only done with very simple toy problems.

**Summary Of The Paper:**

The paper considers the problem of learning a causal model in the POMDP setting. It assumes the learning agent has the ability to collect online experiences through direct interactions with the environment and can access a large collection of offline experiences obtained through the observation of another agent. It further assumes that the observed agent can act based on privileged information hidden from the learning agent. The paper formulates model-based reinforcement learning in this setting as a causal inference problem. The paper then proposes to use offline data as a regularizer during learning. The paper presents empirical results on a number of toy problems.

**Summary Of The Review:**

The setting considered is very limiting as it assumes there are no latent confounders. See the paper for the tricky issues involved:
Shaking the foundations: delusions in sequence models for interaction and control
Pedro A. Ortega, Markus Kunesch, Grégoire Delétang, Tim Genewein, Jordi Grau-Moya, Joel Veness, Jonas Buchli, Jonas Degrave, Bilal Piot, Julien Perolat, Tom Everitt, Corentin Tallec, Emilio Parisotto, Tom Erez, Yutian Chen, Scott Reed, Marcus Hutter, Nando de Freitas, Shane Legg

The experiments are not performed in any non-trivial settings.

The writing makes the paper hard to read. For example, it references rule R2 without specifying where it is or first introducing it.

===
The authors have made their contributions and assumptions more clear, and will add  results comparing with related work. I am happy to upgrade my rating.

---

> ### Author Response · Authors · 2021-11-18
> **Answer**
>
> We thank reviewer 8zjw for his/her review, and summarize his/her concerns as follows:
>
> 1. He/she thinks we assume there is no confounder in the model of the POMDP
> 2. He/she finds the paper is poorly written
> 3. He/she finds the experiments are not performed in any non-trivial setting
>
> Let us now comment in on each of these points.
>
> 1. This point is a major misunderstanding of our paper and we aim to clarify any part of the paper that may have led to this misunderstanding. Indeed, we DO assume that there is confounding in POMDPs, and this is precisely the problem the paper tries to tackle. The issues raised in the pointed paper (Ortega et al. 2021) are precisely the ones we discuss and address in our paper (although we were not aware of the paper at the time of submission, since it only dates from Oct. 20, 2021). To maybe convince you with specific examples, note that their first illustrative example, prize-or-frog, corresponds exactly to our toy problem 1 (see our appendix A.2, page 12 for a guided description of this toy problem). Their “Partially observable, interactive” setting corresponds exactly to the observational regime in our paper, where their “self-delusion problem” is precisely caused by confounding (i.e., the expert has access to privileged information).  The general idea they discuss in their paper, “making sure that actions and observations are regressed as interventions and conditions respectively”, is in essence what our augmented latent model is doing when trained from interventional data (actions) and observational data (observations), with the imposed constraint that $A_t \perp S_t \mid H_t,I=1$ (See figure 1 in our paper). While their paper presents this idea as a concept, we propose a practical solution with theoretical guarantees (in the asymptotic case), backed up with carefully designed experiments. We would be very happy if the reviewer could point us to the parts of our paper that led him/her think there was no confounder, so that we could update these parts and avoid further misunderstanding.
>
> 2. We apologize if you have found the paper poorly written. We’d originally taken great care to introduce the concepts of the paper pedagogically, but unfortunately (due to space limitation) we’ve decided to move some parts of the explanations to the appendix to make room for experimental details, hence the missing reference to rule two of do-calculus (R2). We’ll point to the part of the appendix introducing R2 of do-calculus in the final version of the paper. Regarding your comment “it is not clear how imposing an observational distribution $q(\tau|i = 0)$ acts as a regularizer for the interventional distribution”, it is because it imposes bounds on $q(o_{t+1}|h_t,a_t,i=1)$ (our Theorem 1), the interventional distribution seeked by model-based RL. For a concrete example in a simple bandit setting, we refer you to our guiding example based on our toy problem 1 (A.2, p.12), which is meant to accompany the paper. In the appendix you can also find further explanatory details, such as an introduction to do-calculus (section A.1, p.12), details about our inference procedure (A.3, p.13), and also our complete experimental details and results (A.4, p.14 and A.5, p.15-27 respectively).
>
> 3. We do not agree that our experiments are performed in a trivial setting. While the three toy problems we study in the paper are small-scaled RL problems, the problem we tackle in the paper, confounding in POMDPs, is orthogonal to the problems arising from high-dimensional observation spaces, and is already non-trivial to solve in the low-dimensional setting. Our objective with this paper is to first understand how to solve confounding problems in RL in low dimensions, which is already a non-trivial question, while experiments in higher-dimensional problems are left for future work. Note that our theoretical results are very general and are not limited to low-dimensional problems. We chose on purpose to perform experiments on tabular toy problems, using a also plain, tabular latent-based model, so that the observed behaviours can not be attributed to the (absence of) bells and whistles required for scaling model-based RL to realistic scenarios. We tried to avoid using deep neural networks where hyper-parameter tuning can affect performance greatly, and thereby could reduce the interpretability of the obtained results. Also, adopting such a limited experimental setting allowed us to perform extensive controlled experiments where we could vary the sample sizes and type of confounding present in the data, which are reported in the supplementary material Sections A.5.1 (p.15-21), A.5.2 (p.22-24) and A.5.3 (p.25-27). To conclude, we believe that our experiments are sufficient to support the conclusions of the paper.
>
> We would appreciate if reviewer 8zjw could comment on point 1 above, as there clearly seems to be a misunderstanding of what is the core contribution of the paper, addressing confounding in POMDPs.

---

> > ### Comment · Reviewer_8zjw · 2021-11-30
> > **Thanks for the clarification; more positive now**
> >
> > Thanks for your clarification.
> >
> > Regarding 1 and 2, I think clearly state the assumptions in the context of related work, and connect observation variable and state variable to examples in the paper can help. I noticed that other reviewers also find the assumptions not clear.
> >
> > On 3, I am happy to see that the authors have compared with Kallus et al., in all our confounding settings, and found their approach appears to be significantly more effective. It would have been more helpful if the authors had included the results in the updated paper.
> >
> > I still share some remaining concerns on the contributions and limited experimental results. Even though I think this is a borderline paper, I am happy to upgrade my rating.

---

> > > ### Author Response · Authors · 2021-11-30
> > > **We've updated the paper**
> > >
> > > Thank you for the positive feedback.
> > >
> > > We have updated the paper with the comparison to Kallus et al., after we finished answering the concerns of all the reviewers. You will find these new results if you open the pdf again on page 8, Figure 4. We apologize, we should have notified you of the update.

---

### Official Review · Reviewer_NLhN · 2021-11-02

**Correctness:** 3
**Technical Novelty And Significance:** 2
**Empirical Novelty And Significance:** 2
**Recommendation:** 5
**Confidence:** 3

**Main Review:**

This paper studies the evaluation of interventional distributions in a canonical POMDP model with a finite horizon. The target query is P(o_{t+1} | do(a_{0:t}),o_{0:t}) where A_0, ... A_t represent actions from stage t=0, ... t; O_0, ... O_t+1 represents partially observed states from stage t = 0, ..., t+1. This learning setting is general since it could represent most of treatment regimens in medical domains.

The author first shows that given data collected from interventions, the interventional query P(o_{t+1} | do(a_{0:t}),o_{0:t}) could be consistently estimated using the conditional distribution P(o_{t+1} | a_{0:t},o_{0:t}; I = i) where I represent the intervention policy that generates the data. This is not surprising since the sequential backdoor criterion is entailed in the interventional data. The authors then derive a bound over the product of target query \prod_{t = 0}^{T-1} P(o_{t+1} | do(a_{0:t}),o_{0:t}) from the observational distribution. The result appears interesting at first, but seems to be a simple application of Manski's bound in (Manski, 1989).

The authors validate their results through comprehensive simulations. Results show that estimation using both the observational and interventional data consistently outperforms other learning strategies. However, it is unclear how the combination is done. That is, it would be interesting to see how the authors combine the unbiased estimator from the interventional data with the bound derived from the observational data. Unfortunately, this detail is not elaborated in the main manuscript.

**Summary Of The Paper:**

This paper studies the problems of evaluating interventional distributions (i.e., system dynamics) of a partially observed Markov decision process (POMDP) from samples collected from a combination of randomized experiments and observations of a privileged expert who could access the latent state. The POMDP is presumed to have a finite horizon, e.g., the physician could only perform a finite number of treatments for the same patients. The authors propose an unbiased estimator for evaluating system dynamics from the experimental data. As for the observational distribution where the unobserved confounding exists, the authors derive bounds over unknown system dynamics, estimable from observations.

**Summary Of The Review:**

Overall, the authors study an exciting topic causal identification in POMDP, which is a quite general, and challenging learning setting. My main concern with this paper is its novelty. First, the unbiased estimator in Eq. (4) is not surprising and follows immediately from the backdoor criterion. I am pretty sure many similar MLE estimators have been proposed. Second, the bound in Theorem 1 might be interesting, but appears to be a simple application from the bound in (Manski, 1989). It would be encouraged if the authors could elaborate how to combine these different methods to obtain a more accurate estimation of the target interventional distribution.

---

> ### Author Response · Authors · 2021-11-17
> **Answer [1 / 2]**
>
> We thank reviewer NLhN for his/her comments, and for relating this work to Manski’s bounds which we were not aware of. We also thank reviewer NLhN for acknowledging the importance of causal identification in POMDPs, and how general the setting is.
>
> We note that reviewer NLhN is concerned about the novelty of our proposed approach, in particular because:
> 1. He/she thinks that our main theoretical result (Theorem 1) is a simple application of Manski’s bound
> 2. He/she thinks that our unbiased estimator (Eq. 4) follows immediately from the backdoor criterion, and most likely has already been proposed in the literature
> 3. He/she finds it unclear how Theorem 1 and Eq. 4 are combined to obtain a more accurate estimation of the interventional distribution
>
> Let us now try to convince you that both points 1. and 2. above are wrong, and then elaborate on point 3. which is the main theoretical contribution of the paper.
>
> 1. We disagree, Theorem 1 is much more general than Manski’s bounds. After looking at the pointed paper (Manski, 1989) and following works, we note that Manksi’s bounds only concern interventions on a single variable, within a system of 3 variables. In our framework this corresponds to the contextual bandit setting, i.e., a POMDP with time horizon T=1, and with variables $o_0$, $a_0$ and $o_1$, where the interventional distribution is $p(o_1|do(a_0),o_0)$. In that case then Theorem 1 is indeed a direct application of Manski’s bounds, and similar bounds can already be found in recent literature (Zhang and Bareinboim 2017, 2021). However, Theorem 1 is much more general as it applies to systems with sequential interventions (POMDPs). We actually first derived the bounds of Theorem 1 in the contextual bandit setting (the bounds are equivalent to Manski’s), and it then took us months and lengthy derivations to extend these bounds to the more general POMDP setting. In particular, the difficulty comes from the fact that interventions are nested, and that an intervention in the observational regime $p(o_{t+1}|do(a_t),h_t,i=0)$ is not the same as an intervention in the interventional regime ($p(o_{t+1}|do(a_t),h_t,i=i)$) or even in the mixed regime ($p(o_{t+1}|do(a_t),h_t)$). Manski’s results do not offer a way to handle this additional complexity, which has to be dealt with entirely. The proof of Theorem 1 can be found in our appendix (A.6, p.28), and we hardly see how postulating Manski’s results from the start would simplify our proof. As such, we find that the claim that our result is a simple application of Manski’s bounds is a large overstatement.
>
> 2. We disagree, our estimator does not follow directly from the backdoor criterion. In the interventional regime (DAG in Figure 1), the MLE estimator (eq. 3) directly follows from the backdoor criterion. In the general (augmented) regime however, which mixes both interventional and observational regimes (DAG in Figure 3), the backdoor criterion does not apply. Indeed, there are always non-blocked, non-causal paths which cannot be blocked using the observed variables, such as $A_{t-1} \leftarrow S_{t-1} \to S_{t} \to S_{t+1} \to O_{t+1}$. Our proposed estimator consists in two steps: first learning an augmented model $\hat{q}$ that fits both regimes (eq. 4), and then extracting the learned transition model for the interventional regime ($\hat{q}(o_{t+1}|h_t, a_t, i=1)$). While it is trivial to see that such an estimator is correct (our Proposition 1, which uses the backdoor criterion in the interventional DAG), it is much less trivial to see why this estimator would result in a more accurate model than directly estimating $\hat{q}(o_{t+1}|h_t, a_t, i=1)$ from the interventional data only, which is the main contribution of the paper. Our theoretical demonstration, as to why incorporating observational data in the learning process (eq. 4) is beneficial, does not rely on the backdoor criterion. We try to explain intuitively why this approach is beneficial in the third paragraph on page 6, and then formally in Section 4.4 (Corollary 1). As such, we find that the claim that our estimator follows immediately from the backdoor criterion is wrong, or at the very least misguided. Our main result is not simply that our proposed estimator is unbiased, but also that it leads to a more accurate model due to the use of observational data, for which the backdoor criterion does not hold.

---

> > ### Comment · Reviewer_NLhN · 2021-11-30
> > **Not convinced**
> >
> > > "We disagree, Theorem 1 is much more general than Manski’s bounds. "
> >
> > I am not fully convinced by this statement. Theorem 1 seems to be immediately implied by Manski bound. For convenience, I will use do() operator to represent a distribution induced by policy $i = 1$ and ignore the index $i$ when $i = 0$. Let us take the lower bound in Theorem 1 for $T = 2$ as an example. It states:
> >
> > $P(o_2 | do(a_0, a_1), o_0, o_1) P(o_1 | do(a_0), o_0) \geq P(o_2| a_0, a_1, o_0, o_1) P(a_1|a_0, o_1, o_0)P(o_1|a_0, o_0) P(a_0|o_0)$
> >
> > The above equation implies
> >
> > $P(o_1, o_2|do(a_0, a_1), o_0) \geq P(o_2,o_1,  a_1, a_0 | o_0)$.
> >
> > This coincides with the bounds in (Manski, 1989) by setting treatment $Z = \{A_0, A_1\}$, outcome $Y = \{O_1, O_2\}$ and the context $X = \{O_0\}$.  It is unclear how Theorem 1 improves over the existing partial identification results. Due to this reason, I will keep my current score.

---

> > > ### Author Response · Authors · 2021-11-30
> > > **What about the upper bound ?**
> > >
> > > Thank you so much for taking the time to relate our Theorem 1 to Manski's bounds.
> > >
> > > Indeed, by grouping the variables as you propose it becomes clear that our lower bound is equivalent to Manksi's lower bound.
> > >
> > > However, what about the upper bound ? If we follow your grouping with treatment $Z=A_0,A_1$, outcome $Y=O_1,O_2$ and context $X=O_0$ then it seems to us that Manski's upper bound gives the following:
> > >
> > > $
> > > P(y|do(z),x) \leq P(y,z|x) + 1 - P(z|x) \text{,}
> > > $
> > >
> > > thus
> > >
> > > $
> > > P(o_1,o_2|do(a_0,a_1),o_0) \leq P(a_0,o_1,a_1,o_2|o_0) + 1 - P(a_0,a_1|o_0) \text{.}
> > > $
> > >
> > > is that correct ?
> > >
> > > On the other hand, our upper bound in Theorem 1 is the following:
> > >
> > > $
> > > P(o_1,o_2|do(a_0,a_1),o_0) \leq P(a_0,o_1,a_1,o_2|o_0) + 1 - P(a_0|o_0)P(a_1|o_0,a_0,o_1)
> > > $
> > >
> > > Our bound is tighter. We have $P(a_0|o_0)P(a_1|o_0,a_0,o_1)$ instead of $P(a_0,a_1|o_0)$ on the right-hand side, which is a greater quantity.
> > >
> > > Consider for example a situation where $P(a_0|o_0)=1$, $P(a_1|o_0,a_0,o_1)=1$, $P(o_1|o_0,a_1)=\frac{1}{2}$, $P(o'_1|o_0,a_1)=\frac{1}{2}$ and $P(a_1|o_0,a_0,o'_1)=0$. Then we have
> > >
> > > $
> > > P(a_0|o_0)P(a_1|o_0,a_0,o_1) = 1
> > > $
> > >
> > > (our bound is tight)
> > >
> > > $
> > > P(a_0,a_1|o_0) = P(a_0|o_0)\left(P(o'_1|o_0,a_0)P(a_1|o_0,a_0,o'_1) + P(o_1|o_0,a_0)P(a_1|o_0,a_0,o_1)\right) = \frac{1}{2}
> > > $
> > >
> > > (Manski's bound is not tight)
> > >
> > > We invite you to check whether our reasoning is correct or not. If correct, then we hope you'll be convinced that our Theorem 1 is not a direct application of Manski's bound.

---

> > > > ### Comment · Reviewer_NLhN · 2021-12-01
> > > > **Re: the upper bound**
> > > >
> > > > > On the other hand, our upper bound in Theorem 1 is the following:
> > > > > $P(o_1, o_2|do(a_0, a_1), o_0) \leq P(a_0, o_1, a_1, o_2|o_0) + 1 - P(a_0|o_0)P(a_1|o_0, a_0, o_1)$
> > > >
> > > > This upper bound follows from (1) bounding $P(o_2|do(a_0, a_1), o_0, o_1)$ and $P(o_1|do(a_0), o_0)$ using Manski's bound respectively; and (2) computing the product of the derived Manski's bounds. It is also evident from the derivation that the upper bound in Theorem 2 is not tight.
> > > >
> > > > Overall, I do not dislike the bounding result in Theorem 2, but its contribution appears incremental compared to the existing literature.

---

> > > > > ### Author Response · Authors · 2021-12-02
> > > > > **Thank you for this result. Still, Theorem 1 is not our main contribution.**
> > > > >
> > > > > Thank you again for engaging in discussions, we value this exchange a lot.
> > > > >
> > > > > Just for clarification, there is no Theorem 2 in the paper. We assume you mean Theorem 1 ?
> > > > >
> > > > > You were right, Theorem 1 indeed appears to be an application of Manski's bounds, although additional steps are to be taken. Your two-step procedure (1) then (2) to derive our Theorem 1 using Manski's bounds seems correct, and also we notice that the bounds resulting from the product in step (2) are in general tighter than the bounds in Theorem 1. This is good news to us, as it shows that our learning method can make an even better use of observational data to regularize the recovered causal transition model.
> > > > >
> > > > > Now, we would like to mention three things:
> > > > >
> > > > > 1. The sole purpose of Theorem 1 in our paper is to develop Corollary 1, that is, the observational distribution imposes bounds on the interventional distribution, and thus the observational data acts as a regularizer for the interventional distribution (the causal transition model) that is learned by our method (in the asymptotic regime). How tight these bounds are, and whether they are an incremental result that can easily be derived from Manski's bounds, does not change our argument: our proposed method for merging observational and interventional data in model-based RL is efficient, in the sense that using (enough) observational data results in a better estimator of the causal transition model than when only the interventional data is used.
> > > > >
> > > > > 2. Theorem 1 is not the main contribution of the paper. Our contributions, listed on page 2, are 1) a formulation of model-based RL as a causal inference problem, which is not a trivial thing for the RL community; 2) a generic method for combining offline (observational) and online (interventional) data in model-based RL, which is robust to confounding (correct and efficient); 3) a practical implementation and an experimental evaluation of our method on synthetic toy problems. Note that Theorem 1 is only used for proving the efficiency of our method.
> > > > >
> > > > > 3. The bounds from Theorem 1 are not used within our method, which simply consists in fitting an (augmented) latent-based transition model. This approach closely resembles existing model-based learning procedures for RL (see for example the recent work of Hafner et. al, ICLR 2021). Proposing an approach that looks familiar to the model-based RL community, and is provably able to use confounded data efficiently, will contribute to bridging the gap between the RL and causality communities. This is the main contribution of the paper, not the bounds in Theorem 1.
> > > > >
> > > > > We are happy that you've related our paper to existing works in the causal inference literature, and in particular to Manski's bounds. We are also grateful that you've showed us how tighter bounds can be obtained by applying the results from (Manski, 1989) sequentially. We therefore propose to acknowledge (Manski, 1989) better in the paper, and to mention explicitly that our Theorem 1 can be shown to be a consequence of Manski's bounds, and that a tighter version exists. We would happily reformulate Theorem 1 as a Corollary based on Manski's bounds, and also derive tighter bounds from the procedure you've pointed us to, however that would maybe make too big of a change to the paper at this point, given that the reviewers could not assess these changes before the final version.
> > > > >
> > > > > We thank you again for your valuable time, and hope you'll take the above points into consideration in your final assessment of the paper.

---

> ### Author Response · Authors · 2021-11-17
> **Answer [2 / 2]**
>
> 3. It seems the explanations found in the paper did not allow reviewer NLhN to fully grasp how our proposed approach works, and thus how Theorem 1 relates to it. Let us now try to bring some clarifications, which we will gladly add to the paper as well. A key point is that our approach consists in training a latent model that fits both observational and interventional data (eq. 4), and then retaining only the resulting interventional transition model. In details: 1) we train a single latent model that fits data from both the observational and interventional regimes, by sharing the same latent space $\mathcal{Z}$ and the same building components $q(z_0)$, $q(o_t|z_t)$ and $q(z_{t+1}|z_t,a_t)$. Only one component, $q(a_t|h_t,z_t,i)$, allows for some flexibility between the interventional and observational distributions, and thus allows the model to distinguish both regimes; 2) we show that when our model fits the observational distribution perfectly (a sufficient condition is infinite observational data + large enough state space $\mathcal{Z}$), then the model $\hat{q}$ learned in (eq. 4) entails non-trivial bounds on the interventional transition model (Theorem 1). As a consequence, in the asymptotic observational scenario (large enough observational data set), we show that our interventional transition model estimator, obtained from interventional and observational data using eq. 4, generalizes better than the one obtained without observational data (Corollary 1). We hope that this explanation, in addition to our answers to points 1. and 2. above, will bring light to how Theorem 1 relates to our proposed approach.
>
> We thank reviewer NLhN again for his/her comments, and we hope that he/she will find the requested clarification in our answer. We will discuss how our method relates to Manski’s bounds in our related work section in the final version of the paper. Thanks again for pointing us to this very relevant seminal work. We will also gladly work on improving the parts of the paper which have been confusing, if reviewer NLhN would point us to the specific parts of the paper that are unclear to him/her.
>
> Finally, we would also like to emphasize that the bounds derived in Theorem 1 are not the only contribution of the paper. Relating those bounds to the generalization capacity of an augmented latent variable-based model, and then relating this model to model-based reinforcement learning, within a general POMDP framework that mixes confounded and unconfounded data, is also a contribution of the paper. This contribution holds regardless of whether those bounds appear to be a simple application of a previous result or not.

---

### Official Review · Reviewer_cYgx · 2021-11-03

**Correctness:** 3
**Technical Novelty And Significance:** 3
**Empirical Novelty And Significance:** 3
**Recommendation:** 5
**Confidence:** 3

**Main Review:**

**Strengths**

1. This paper addressed an interesting and important question in RL, i.e., how to use the observational data to improve the performance of online learning.

2. The authors claimed that their setting is non-trivial by considering the unobserved confounders in the observational data.

3. Their method was shown to be valid and promising both theoretically and empirically.

**Weaknesses**

1. Since one major contribution claimed in this paper is to bridge the causal inference with reinforcement learning, I was expecting that the authors could use a more rigorous causal framework and necessary assumptions to ensure the validation of their method and theory. For instance, to replace the do-operator with the conditional probabilities, one should assume ignorability or exogeneity. Please refer to [1] below and add related assumptions.

[1] Pearl, Judea. "Models, reasoning and inference." Cambridge, UK: Cambridge university press 19 (2000).

2. I am not very convinced why online data particularly follows POMDP and offline data follows (possibly) privileged POMDP. Does a pre-testing procedure is required to justify the model assumptions?

3. There are at least two directions of literature that the authors should pay attention to and justify their novelty.

(a). First, a number of works have proposed to combine observational and experimental data (though not for RL), such as [2], [3], etc. The authors may justify why they use the augmentation procedure and will this procedure achieve the usually desired doubly robust property?

[2] Athey, Susan, Raj Chetty, and Guido Imbens. "Combining experimental and observational data to estimate treatment effects on long term outcomes." arXiv preprint arXiv:2006.09676 (2020).

[3] Cooper, Gregory F., and Changwon Yoo. "Causal discovery from a mixture of experimental and observational data." arXiv preprint arXiv:1301.6686 (2013).

(b). There are increasing works regarding combining offline and online data in RL, while it seems that the authors only discussed partial of them. See some works below.

[4] Nair, Ashvin, et al. "Accelerating online reinforcement learning with offline datasets." arXiv preprint arXiv:2006.09359 (2020).

[5] Gelly, Sylvain, and David Silver. "Combining online and offline knowledge in UCT." Proceedings of the 24th international conference on Machine learning. 2007.

4. I don't agree with the statement by the authors that 'Although we would have loved to compare against those approaches, the lack of available code did prevent us from running a fair comparison.' Actually, by searching these cited papers' titles with 'GitHub', I did find their implementations (as follows). Thus, the authors should add the comparison studies to justify their better performance.

[6] Nathan Kallus, Aahlad Manas Puli, Uri Shalit. Removing Hidden Confounding by Experimental Grounding. NIPS 2018.

https://github.com/CausalML/RemovingHiddenConfounding

[7] Elias Bareinboim, Andrew Forney, and Judea Pearl. Bandits with unobserved confounders: A causal approach. In NIPS, 2015.

https://github.com/nanavatirutu/CausalBandits



**Summary Of The Paper:**

This paper considers the model-based reinforcement learning (RL) problem by combining the offline and online data. The online data, i.e., the interventional data, is generated from the standard partially-observable Markov decision process (POMDP), while the offline data, i.e., the observational data, is generated from the privileged POMDP where the offline learner had the access to the state information (i.e., the unobserved confounder) to make an action. The authors proposed an augmented learning procedure to safely combine these two separate different sources and learn a more efficient policy. Their method is shown both theoretically and empirically better than not using offline data. My main concerns lie in their framework and assumptions, novelty compared with existing literature, and comparison studies.

**Summary Of The Review:**

I think this is a borderline paper that addressed an important question with reasonably good performance while lacking necessary elaboration and justification. As commented in my 'Main Review', my major concerns to recommend this paper lie in their framework, novelty compared with existing literature, and comparison studies. I am willing to upgrade if my concerns can be addressed during the rebuttal period.

---

> ### Author Response · Authors · 2021-11-19
> **Answer [1/3]**
>
> We thank reviewer cYgx for his/her comments, and we are happy that he/she identified well our objective to bridge Pearl’s causality framework and RL, and recognizes the importance of the problem addressed in the paper. We summarize his/her concerns as follows:
>
> 1. He/she thinks that our causal framework is not rigorous, and believes some necessary assumptions are missing from the paper, such as ignorability or exogeneity.
> 2. He/she is not convinced how online / offline data respectively follow a standard POMDP / privileged POMDP, and wonders if a pre-testing procedure is required.
> 3. He/she questions the novelty of our approach, and points us to missing references.
> 4. He/she requests a justification for our augmentation procedure, and asks if this procedure achieves the usually desired doubly robust property.
> 5. He/she requests a comparison to competing approaches, and points us to publicly available codes.
>
> Let us now comment on each of these points.
>
> 1. We follow the causal framework of Judea Pearl, which is standard and we believe is rigorous. From Pearl’s textbook which you point us to, the two concepts you mention, ignorability and exogeneity, are shown to be equivalent to the concept of unconfoundedness (“no confounding”). See, e.g., Sections 5.4.3 and 7.4.5 of the book. We do discuss at great length the “no confounding” assumption (and thus ignorability and exogeneity) in Section 3, and we do not mention it as a necessary condition to our approach simply because our approach does not make that assumption. See our problem statement (Section 4.1), which explicitly mentions that confounding is allowed in the observational regime.
> > We consider a generic situation where two datasets of POMDP trajectories [...] are available, sampled respectively in the interventional regime [...] and in the observational (potentially confounded) regime [...].
> No further assumption is made about the data, other than the fact that we restrict ourselves to discrete spaces and finite-horizon POMDPs (Section 2.2). We find it useful that you raised the question of ignorability and exogeneity, and we will add a paragraph in the appendix to further clarify this point.
>
> 2. Online data follows a standard POMDP regime by definition. The decision-making policy is known to the learning agent, and relies only on historical observations (and actions). Because the policy does not use any privileged information, there is no confounding (see the last paragraph of Section 3.1). Offline data, on the other hand, can either follow the standard POMDP regime or the privileged POMDP regime, depending on the definition. If the offline data comes from a policy that we know relies only on historical observations (and actions), then it also falls into the standard POMDP regime. This is the case for example if the offline data also comes from the learning agent, when it was using a different policy than the current one. In the paper we refer to offline data in the broadest sense, when the decision-making policy is completely unknown, and the assumption that the offline policy relies only on observed historical information can not be guaranteed. This is the case, for example, when the offline data comes from a human agent, which may rely on additional information that is not observed by the learning agent. The standard POMDP regime then can not be assumed, and one must consider the more general privileged POMDP regime, where the privileged agent can have access to the entire hidden state (thus allowing the worst-case confounding scenario). This situation is motivated in the introduction, with two concrete examples in self-driving cars and in medicine, and is also discussed at the end of Sections 3.1 and 3.2.
> A pre-testing is not required to assert our model assumptions. Our only assumption about the data is that the interventional data follows a standard POMDP regime, which is guaranteed if the data is collected from the learning agent through interactions, while the observational data relaxes that assumption and allows for any kind of confounding. As a convincing piece of evidence, note that we evaluate our method experimentally under various types of confounding in the observational data, and that it behaves nicely in all situations, even when there is no confounding at all (the random scenario in Figure 5).

---

> ### Author Response · Authors · 2021-11-19
> **Answer [2/3]**
>
> 3. Thank you for pointing us to these papers. Our approach is novel with respect to the references pointed to us, as none of these papers addresses the problem of combining unconfounded (interventional) and confounded (observational) data in RL. References [2,3] indeed belong to the large body of existing works that combine observational and observational data to solve causal inference problems. However, after a closer look these papers appear rather distant from our own contribution, as they both address different questions that the one we raise in our paper, and none of them can be applied to RL. We believe papers [2-3] would better deserve to be discussed in a review paper on causal inference, which is not our intent with this paper. Papers [4,5] both combine offline and online data in RL, however the problems they try to address do not relate to confounding. Still, we agree that they could be worth being discussed under the perspective of our causal framework, and we will include them in our related work section. Let us now discuss each of these papers in detail.
> Paper [2] considers the use of observational and interventional data for a causal inference task, however the setup and the problem they address is quite different from ours, and is not directly applicable to RL (or even bandit) problems. The authors consider a specific bandit setting with two outcome variables, primary and secondary, and ask if a causal model for the primary outcome can be learned given that the primary outcome is only observed in the observational (confounded) regime. They propose a method that requires a specific condition, that the confounding bias on the primary outcome can be deduced from the confounding bias on the secondary outcome. This result is interesting, however it is not applicable to regular RL (or even bandit) problems, and can not be directly compared to ours.
> Paper [3] considers the use of observational and interventional data for causal structure learning, which is very interesting but again is quite a different problem than the one we discuss in our paper, which is estimating a causal effect given that the causal structure is known (a POMDP).
> Paper [4] addresses the bootstrapping error problem that arises with off-policy data, and which is due to bootstrapping from actions that lie outside of the data distribution [see Kumar et al., NeurIPS 19]. This problem is again quite different from the one we address in our paper, confounding, and is orthogonal since bootstrapping errors can happen without confounding, and vice-versa.
> Paper [5] addresses the problem of combining online and offline data in an RL algorithm, however it assumes that there is no hidden variable (no confounding). Thus, their setting falls into a purely interventional regime, and is trivial with regard to the problem of causal identification.
>
> 4. Our augmented learning procedure is justified in the sense that 1) it results in a more robust estimator of the causal transition model seeked in RL (Theorem 1 and Corollary 1, also our experimental results), and 2) it closely resembles existing model-based learning procedures for RL (see for example the recent work of Hafner et. al, ICLR 2021). Maybe our whole augmented learning procedure, which requires to fit a latent-variable model, is not necessary to correct for confounding in RL (see our comments on model-free RL in the conclusion), however we believe that proposing an approach that looks familiar to the model-based RL community, and is provably able to correct confounding, will contribute to bridging the gap between the RL and causality communities.
> Regarding your comment about whether our method achieves the doubly robust property, this question is interesting but is out of the scope of this paper. From what we understand, the doubly robust property allows to consistently estimate a causal effect even in the presence of (partial) model misspecification. In our paper we assume perfect model specification, that is, our model family $Q$ has enough capacity to model the desired causal model, which is a common assumption in RL and in deep learning. The question of how our method behaves in the presence of model misspecification is interesting and would be worth studying in a future work, but is not the scope of this paper.

---

> ### Author Response · Authors · 2021-11-19
> **Answer [3/3]**
>
> 5. Thank you for pointing us to these existing codes. It appears we have not been very rigorous in our search. We were able to evaluate the approach from Kallus et al. on our bandit problem (it does not apply to general POMDPs), and we will include the results in the main paper as well as in the appendix. It appears that our approach is significantly more effective than Kallus et al., in all our confounding settings, which brings an additional piece of evidence that supports the competitiveness of our method.
> Regarding the second code you mention however (the Bareinboim, Forney and Pearl paper), a comparison would not be fair nor interpretable, as the approach proposed in this paper is for a different RL setting, when the learning agent has access to the privileged agent’s preference (see the casino example in their paper, and also our discussion it in our related work section). Therefore, this approach can not be included in our comparative experiments.
>
> We thank reviewer cYgx again for his/her comments which will help us improve our paper, and we hope that we have addressed all of his/her concerns properly. Please do not hesitate to give us feedback and engage in further discussions.

---

### Official Review · Reviewer_Xuy7 · 2021-11-03

**Correctness:** 3
**Technical Novelty And Significance:** 2
**Empirical Novelty And Significance:** 2
**Recommendation:** 5
**Confidence:** 4

**Main Review:**

My main concerns are on the experimental side.

- The results on the three very low-dimensional synthetic toy problems are quite limited. It is hard to judge the validity of the proposed method. As we know, the RL problems would become exponentially difficult as the state dimension increases. Also, in the current RL community most RL algorithms take image pixels as input. Without such experiments, it is unclear how the proposed method work in the real world scenarios.

- Comparing with several baselines and SOTA methods is an important way to demonstrate the superiority of the proposed approach. Unfortunately, such a comparison is missing in the paper. I suggest the authors should add some, which would make the paper more convincing. E.g., Rezende et al. (2020), Kallus et al. (2018), Zhang&Bareinboim (2020), etc.

**Summary Of The Paper:**

The authors study the POMDP problem from the causal perspective, and the propose to combine offline and online data to infer the transition model via deconfounding. On the theoretical side, they show that the proposed method is correct and efficient in terms of generalization guarantees. On the experimental side, they evaluate the proposed method on three synthetic toy problems.

**Summary Of The Review:**

The experimental results are quite limited so that they are not enough to support the claims in the paper.

---

> ### Author Response · Authors · 2021-11-20
> **Answer [1/2]**
>
> We thank reviewer Xuy7 for his/her comments. We note that reviewer Xuy7 did not question the novelty of our work or the importance of the problem it tries to address, but remains concerned about the following points:
>
> 1. He/she finds the paper is missing experiments on high-dimensional problems (e.g., with image pixels) to judge the validity of our method, and questions whether it works in real-world scenarios.
>
> 2. He/she finds the paper is missing an experimental comparison to state-of-the-art methods, such as Rezende et al. (2020), Kallus et al. (2018), or Zhang and Bareinboim (2020).
>
> 3. As a result, he/she finds our experimental results are too limited to support the claims of the paper.
>
> Let us now comment on each of these points.
>
> 1. We respectfully disagree on the premise that RL contributions must necessarily be evaluated on large-scale problems. We would like to put forward three arguments: 1) the problem we address in the paper, confounding in RL, is orthogonal to the issues that arise with high-dimensional spaces, and is already non-trivial to solve in the low-dimensional setting. See also our answer to reviewer 8zjw, point 3. 2) it is common for theoretical RL papers to perform experiments on low-dimensional toy problems, and the abstract domains we consider are standard tools for theoretical papers involving POMDPs. The tiger problem for example is used in the following recent papers: Katt et al, ICML’17, Learning in POMDPs with Monte Carlo Tree Search; Subramamian et al., CDC’19, Approximate information state for partially observed systems; Cui and Khardon, NeurIPS’19, Sampling Networks and Aggregate Simulation for Online POMDP Planning; Alt et al., NeurIPS’20, POMDPs in Continuous Time and Discrete Spaces. 3) there exist important real-world domains (e.g., in the medical domain) that do not involve pixel-based or high-dimensional spaces. See, e.g., the comments of reviewer NLhN, who acknowledges that our learning setting is general and could represent most treatment regimes in medical domains.
>
> 2. Acknowledging and comparing to existing works is indeed essential, and the reason why we did not include Kallus et al. (2018) or Zhang and Bareinboim (2020) was simply because we could not find an official implementation for these papers. Thanks to reviewer cYgx who pointed us to the implementation of Kallus et al., we were able to evaluate that approach on our bandit problem (it does not apply to general POMDPs), and we will include the results in the main paper as well as in the appendix. It appears that our approach is significantly more effective than Kallus et al., in all our confounding settings, which brings an additional piece of evidence that supports the competitiveness of our method. Regarding a comparison to Rezende et al. (2020), “Causally Correct Partial Models for Reinforcement Learning”, their approach applies to a purely observational regime (off-policy evaluation), and does not fit our problem setup which mixes both interventional and observational data.

---

> ### Author Response · Authors · 2021-11-20
> **Answer [2/2]**
>
> 3. We respectfully disagree. The claims of the paper (p. 2) are the following:
>  - formalizing model-based RL as a causal inference problem
>  - proposing a generic, provably correct and provably efficient method in the presence of interventional and observational (potentially confounded) data
>  - proposing a practical implementation, which is effective on three synthetic toy problems
>
> We do not claim to propose an implementation that readily scales to high-dimensional RL problems. To clarify this point we propose to update our conclusion and make this limitation more explicit, from
> >  One limitation of our method is that it adds an additional challenge on top of model-based RL, that of learning a latent-based transition model.
>
> to
>
> > One limitation of our method is that it adds an additional challenge on top of model-based RL, that of learning a latent-based transition model, which can become problematic in high-dimensional RL settings.
>
> Although evaluating our method (or proposing a variant of it that works) in high-dimensional settings is a direction we would like to pursue (see our discussion of Hafner et al., ICLR 2021 in the conclusion), it also involves addressing additional challenges which are out of the scope of this paper. The confounding problem is non-trivial to solve even in simple RL settings, and to convince you we would like to point out that in our updated experiments the SOTA method from Kallus et al. is unsuccessful even in a simple binary bandit problem. We demonstrate in our experiments the robustness of our method to different degrees of confounding, and we even perform experiments in the extreme scenario where there is no confounding at all. Therefore, we do believe that our experiments are sufficient to support the claims of the paper.
>
> We hope that these arguments, combined with our updated experimental results with Kallus et al. (2018), will convince reviewer Xuy7 of the value of this paper to the ICLR community, even without experiments in large-scale RL problems.

---

### Author Response · Authors · 2021-11-20
**Discussions**

The authors thank again the reviewers for their time and valuable reviews. In addition, we would like to highlight that the discussion period will end on Monday.

We hope that we have covered the reviewers’ concerns. It would be helpful to know their reaction to our responses while there is still time to engage in discussion.

The authors would really appreciate it if the reviewers could comment on the authors’ responses and raise any remaining concerns.

---

### Author Response · Authors · 2021-11-30
**Updated manuscript**

Dear reviewers, we have uploaded an new version of the manuscript that takes into account your comments. In particular, there is now a comparison to the method from Kallus et al. in the bandit experiment (door toy problem), which can be found on page 8, Figure 4.

---

### Decision · Program_Chairs · 2022-01-20

**Decision:**

Reject

**Comment:**

In this paper, the authors provide a model-based approach for combining experimental and observational data in reinforcement learning, specifically in POMDPs.

The paper was not received very favorably by reviewers, with the main concerns revolving around: (a) writing quality, (b) validation, (c) extent of contribution given existing work on causal RL.

In preparing your revision, in addition to clarifying writing, and adding better validation, I would urge the authors to consult existing causal inference literature on point and partial identification in settings related to RL, such as off-line policy learning.  This will help address issues of novelty by extending their approach to settings with more types of confounding.  In addition to useful references suggested by reviewers, another useful draft may be:

"Path-Dependent Structural Equation Models." Srinivasan, R., Lee, J., Bhattacharya, R., and Shpitser, I.. In Proceedings of the Thirty Seventh Conference on Uncertainty in Artificial Intelligence.